# Repression of SMAD3 by STAT3 and c-Ski induces conventional dendritic cell differentiation

Jeong-Hwan Yoon[1,2,3,4], Eunjin Bae[2,5,14], Yasuo Nagafuchi[6], Katsuko Sudo[7], Jin Soo Han[4], Seok Hee Park[8], Susumu Nakae[9], Tadashi Yamashita[10], Ji Hyeon Ju[11], Isao Matsumoto[12], Takayuki Sumida[12], Keiji Miyazawa[13], Mitsuyasu Kato[14], Masahiko Kuroda[2], In-Kyu Lee[1], Keishi Fujio[6], Mizuko Mamura[1,3,15]

A pleiotropic immunoregulatory cytokine, TGF-β, signals via the receptor-regulated SMADs: SMAD2 and SMAD3, which are constitutively expressed in normal cells. Here, we show that selective repression of SMAD3 induces cDC differentiation from the CD115[+] common DC progenitor (CDP). SMAD3 was expressed in haematopoietic cells including the macrophage DC progenitor. However, SMAD3 was specifically down-regulated in CD115[+] CDPs, SiglecH[−] pre-DCs, and cDCs, whereas SMAD2 remained constitutive. SMAD3-deficient mice showed a significant increase in cDCs, SiglecH[−] pre-DCs, and CD115[+] CDPs compared with the littermate control. SMAD3 repressed the mRNA expression of FLT3 and the cDC-related genes: IRF4 and ID2. We found that one of the SMAD transcriptional corepressors, c-SKI, cooperated with phosphorylated STAT3 at Y705 and S727 to repress the transcription of SMAD3 to induce cDC differentiation. These data indicate that STAT3 and c-Ski induce cDC differentiation by repressing SMAD3: the repressor of the cDC-related genes during the developmental stage between the macrophage DC progenitor and CD115[+] CDP.

## Introduction

Conventional DCs (cDCs) are highly potent antigen-presenting cells, which initiate and orchestrate adaptive immunity (Steinman, 2012), whereas plasmacytoid DCs (pDCs) detect pathogen-derived nucleic acids, thereby producing type I interferon upon viral infection (Reizis, 2019). To define cell lineages, intensive efforts have been devoted to elucidate the mechanisms whereby these two major DC lineages develop from their haematopoietic progenitor cells. A network of cytokine signalling pathways and transcription programmes control development of DC subsets from distinct haematopoietic lineages (Lin) (Belz & Nutt, 2012; Miller et al, 2012; Merad et al, 2013; Murphy et al, 2016; Dress et al, 2018, 2019; Rodrigues et al, 2018; Nutt & Chopin, 2020; Anderson et al, 2021; Cabeza-Cabrerizo et al, 2021; Ginhoux et al, 2022; Zhang et al, 2023). Several cytokine receptors such as Fms-related tyrosine kinase 3 (FLT3; CD135), c-Kit (CD117), and macrophage colony-stimulating factor receptor (M-CSFR; CD115) are the markers to distinguish Lin[−] DC progenitors in mouse BM. Common myeloid progenitors, common lymphoid progenitors, and lymphoid-primed multipotent progenitors (LMPPs) are the early DC progenitors, which differentiate into the intermediate progenitor, macrophage DC progenitor (MDP). Downstream of the MDP is the common DC progenitors (CDPs) comprised of CD115[+] and CD115[−] CDPs, which give rise to conventional/classical DCs (cDCs) and plasmacytoid DCs (pDCs), respectively, in the steady-state condition (Onai et al, 2007; Schraml et al, 2015; Onai & Ohteki, 2016; Anderson et al, 2021; Cabeza-Cabrerizo et al, 2021; Ginhoux et al, 2022). Murine CD115[+] CDPs have been reported as DC progenitors with major cDC differentiation potential (Onai et al, 2007; Schraml et al, 2013). CDPs differentiate into Lin[−]CD11c[+]MHCII[−]CD135[+]CD172α[−] pre-DCs divided into four subsets based on the expression patterns of sialic acid–binding Ig-like lectin (Siglec)-H (SiglecH) and Ly6C (Schlitzer et al, 2015).

Regarding the cDC-restricted progenitors, pre-cDCs were defined as a CD11c[+]MHCII[−] proliferative precursor in BM and lymphoid tissues (Diao et al, 2006; Naik et al, 2006). A single-cell analysis identified SiglecH[+]Ly6C[+] pre-DCs as cDC-restricted precursors (Schlitzer et al, 2015). Comprehensive studies have identified key transcription factors regulating specification and differentiation of cDC subsets; the Ets-family transcription factor PU.1 is essential for cDC differentiation through inducing

[1]Biomedical Research Institute, Kyungpook National University Hospital, Daegu, Republic of Korea [2]Department of Molecular Pathology, Tokyo Medical University, Tokyo, Japan [3]Shin-Young Medical Institute, Chiba, Japan [4]Institute for the 3Rs, Department of Laboratory Animal Medicine, College of Veterinary Medicine, Konkuk University, Seoul, Republic of Korea [5]Department of Companion Health, Yeonsung University, Anyang, Republic of Korea [6]Department of Allergy and Rheumatology, Graduate School of Medicine, The University of Tokyo, Tokyo, Japan [7]Animal Research Center, Tokyo Medical University, Tokyo, Japan [8]Department of Biological Sciences, Sungkyunkwan University, Suwon, Republic of Korea [9]Graduate School of Integrated Sciences for Life, Hiroshima University, Hiroshima, Japan [10]Laboratory of Veterinary Biochemistry, Azabu University School of Veterinary Medicine, Sagamihara, Japan [11]Department of Rheumatology, Catholic University of Korea, Seoul St. Mary Hospital, Seoul, Republic of Korea [12]Department of Internal Medicine, University of Tsukuba, Tsukuba, Japan [13]Departments of Biochemistry, University of Yamanashi, Yamanashi, Japan [14]Department of Experimental Pathology, Graduate School of Comprehensive Human Sciences and Faculty of Medicine, University of Tsukuba, Tsukuba, Japan [15]Department of Advanced Nucleic Acid Medicine, Tokyo Medical University, Tokyo, Japan

Correspondence: mikoeyo@gmail.com

DC-SCRIPT while repressing the pDC-related genes (Chopin et al, 2019); the helix–loop–helix transcription factor, an inhibitor of DNA-binding protein 2 (ID2), is required for the development of splenic CD8α⁺ DC subset and Langerhans cells (Hacker et al, 2003); interferon regulatory factors (IRF)-2, IRF-4, and IRF-8 regulate cDC and pDC differentiation (Merad et al, 2013); ID2 and E2-2 induce cDCs and pDCs, respectively, with mutual antagonism (Ghosh et al, 2010); and STAT3 is required for FLT3-dependent DC differentiation (Laouar et al, 2003).

TGF-$\beta$ is a pivotal cytokine to regulate haematopoiesis and immune cell development in a pleiotropic manner (Söderberg et al, 2009; Challen et al, 2010; Blank & Karlsson, 2015; Sanjabi et al, 2017). TGF-$\beta$ has been reported to exert the multifaceted effects on DC differentiation depending on the developmental stages (Seeger et al, 2015). TGF-$\beta$ promotes DC development from CD34⁺ haematopoietic progenitors (Strobl et al, 1996; Riedl et al, 1997). TGF-$\beta$1 is required for immature DC development, whereas it blocks DC maturation (Yamaguchi et al, 1997). TGF-$\beta$1 directs differentiation of CDPs into cDCs by inducing cDC instructive factors, IRF4 and RelB, and ID2 (Felker et al, 2010). TGF-$\beta$1 induces DC-associated genes such as *Flt3*, *Irf4*, and *Irf8* in multipotent progenitors at the steady state (Sere et al, 2012). Signalling mechanisms underlying pleiotropic functions of TGF-$\beta$ have been vigorously investigated. The canonical TGF-$\beta$ signalling pathway is initiated by ligand-bound activated TGF-$\beta$ type I receptor (T$\beta$RI)/phosphorylated TGF-$\beta$ receptor–regulated SMADs (R-SMADs): SMAD2 and SMAD3. In spite of their high homology, SMAD2 and SMAD3 exert differential functions depending on the context via mechanisms yet to be fully determined (Brown et al, 2007; Heldin & Moustakas, 2012; Batlle & Massague, 2019; Miyazawa et al, 2024).

Here, we report that one of the R-SMADs, SMAD3, is specifically repressed in cDCs, SiglecH⁻ pre-DCs, and CD115⁺ CDPs. SMAD3 is the repressor of the transcription factors essential for cDC differentiation such as FLT3, ID2, and IRF4. We have discovered that transcription of the *Smad3* gene is repressed by STAT3 in cooperation with c-SKI, one of the SKI/SNO proto-oncoproteins that inhibit TGF-$\beta$ signalling as the transcriptional corepressor of the SMAD proteins (Deheuninck & Luo, 2009; Batlle & Massague, 2019) for cDC differentiation. Our findings suggest that down-regulation of SMAD3 is required for cDC differentiation downstream of CD115⁺ CDPs.

## Results

### Selective down-regulation of SMAD3 in cDCs

To examine the expression of SMAD2 and SMAD3 in cDCs and their progenitors, we isolated LMPPs as Lin⁻Sca-1⁺CD34⁺CD117⁺CD135⁺ cells; the DC progenitor cells: MDPs as Lin⁻CD117ʰⁱCD135⁺CD115⁺Sca-1⁻, and CD115⁺ CDPs as Lin⁻CD117ⁱⁿᵗCD135⁺CD115⁺CD127⁻ (Fogg et al, 2006; Onai & Ohteki, 2016), SiglecH⁻Ly6C⁻/SiglecH⁻Ly6C⁺CD11c⁺MHCII⁻CD135⁺CD172α⁻ pre-DCs (Schlitzer et al, 2015), and differentiated cDCs: BM CD11b⁺CD11c⁺ cDCs and splenic CD11b⁺CD11c⁺ cDCs of C57BL/6 mice. We also examined their expression patterns in BM-derived DCs (BMDCs): GM-CSF plus IL-4–induced BMDCs that yield CD11b⁺ cDCs

and FLT3L-induced BMDCs that yield both cDCs and pDCs (Yamaguchi et al, 1997; Waskow et al, 2008). SMAD2 mRNA was expressed in all examined cells, whereas SMAD3 mRNA was reduced to almost undetectable levels in CD115⁺ CDPs, SiglecH⁻ pre-DCs, CD11b⁺CD11c⁺ cDCs, and CD11cʰⁱ BMDCs (Fig 1A). Immunoblotting confirmed that the SMAD2 protein (60 kD) is kept expressed, whereas the SMAD3 protein (52 kD) expressed in whole BM was reduced to an undetectable level in GM-CSF plus IL-4–induced BMDCs (Fig S1A). Immunocytochemistry determined by the proximity ligation assay (PLA; Söderberg et al, 2006) showed that the SMAD3 protein was not detected in CD115⁺ CDPs, SiglecH⁻ pre-DCs, and CD11c⁺ cells, whereas the SMAD2 protein was expressed in all examined subsets (Fig 1B and C). cDCs are further classified into type 1 cDCs (cDC1s) and type 2 cDCs (cDC2s) with distinct features and functions (Murphy et al, 2016; Ginhoux et al, 2022; Zhang et al, 2023). We confirmed that sorted CD8a⁺ (cDC1s) or CD11b⁺ (cDC2s) cells out of CD11c⁺ cells were both negative for SMAD3 (Fig 1B and C).

The expression of SMADs is generally ubiquitous and constitutive in normal cells (Brown et al, 2007; Miyazawa et al, 2024). We confirmed that SMAD2 and SMAD3 proteins were expressed in macrophages, naïve and activated CD4⁺ T cells (Fig S1B).

We screened the expression patterns of SMAD2 and SMAD3 in immune cells using representative open public data sources (ImmGen consortium: Heng et al, 2008; Yoshida et al, 2019; UCSC Cell Atlas: Domínguez Conde et al, 2022). Microarray (Fig S2A) and RNA-seq (Fig S2B) of murine immune cells and their precursor cells, as well as human single-cell RNA-seq (Fig S2C), show that the expression of SMAD3 mRNA is specifically down-regulated in mouse and human cDC subsets, whereas SMAD2 mRNA is expressed in all immune cells.

Our data validated by the open public data sources show that SMAD3 is selectively and specifically down-regulated in cDCs and its progenitors: CD115⁺ CDPs and SiglecH⁻ pre-DCs.

### SMAD3 deficiency enhances cDC differentiation between the MDP and CD115⁺ CDP in vivo

We next examined the roles of SMAD3 in cDC differentiation in vivo using *Smad3⁻/⁻* mice. Immunophenotyping was performed using flow cytometry according to the published gating procedures (Liu et al, 2020; Fig S3A). The numbers of cDC subsets: CD11c⁺, CD11b⁺CD11c⁺, and CD8⁺CD11c⁺ cells in BM, spleens, and superficial and mesenteric lymph nodes of *Smad3⁻/⁻* mice were significantly increased compared with those of littermate control *Smad3⁺/⁺* mice (Fig 2A).

We next traced upstream cDC progenitors. A significantly increased CD115⁺ CDP (Lin⁻CD117ⁱⁿᵗ) along with a decreased MDP (Lin⁻CD117ʰⁱ) was observed in the BM of *Smad3⁻/⁻* mice compared with that of the littermate control (Fig 2B). DNGR-1 (encoded by the *Clec9a* gene and known as CLEC9A and CD370)–positive CDPs are cDC-restricted (Schraml et al, 2013; Cabeza-Cabrerizo et al, 2021). CX3CR1 is expressed on the MDP and cDC-P (Liu et al, 2009). Expressions of *Cx3cr1* and *Dngr1* mRNA in Lin⁻CD115⁺ BM cells and the proportion of CX3CR1⁺CD370⁺Lin⁻CD117ⁱⁿᵗCD135⁺CD115⁺ BM cells were significantly increased in *Smad3⁻/⁻* mice compared with the littermate control (Fig 2C). SiglecH⁻Ly6C⁺ and SiglecH⁻Ly6C⁻ pre-DCs

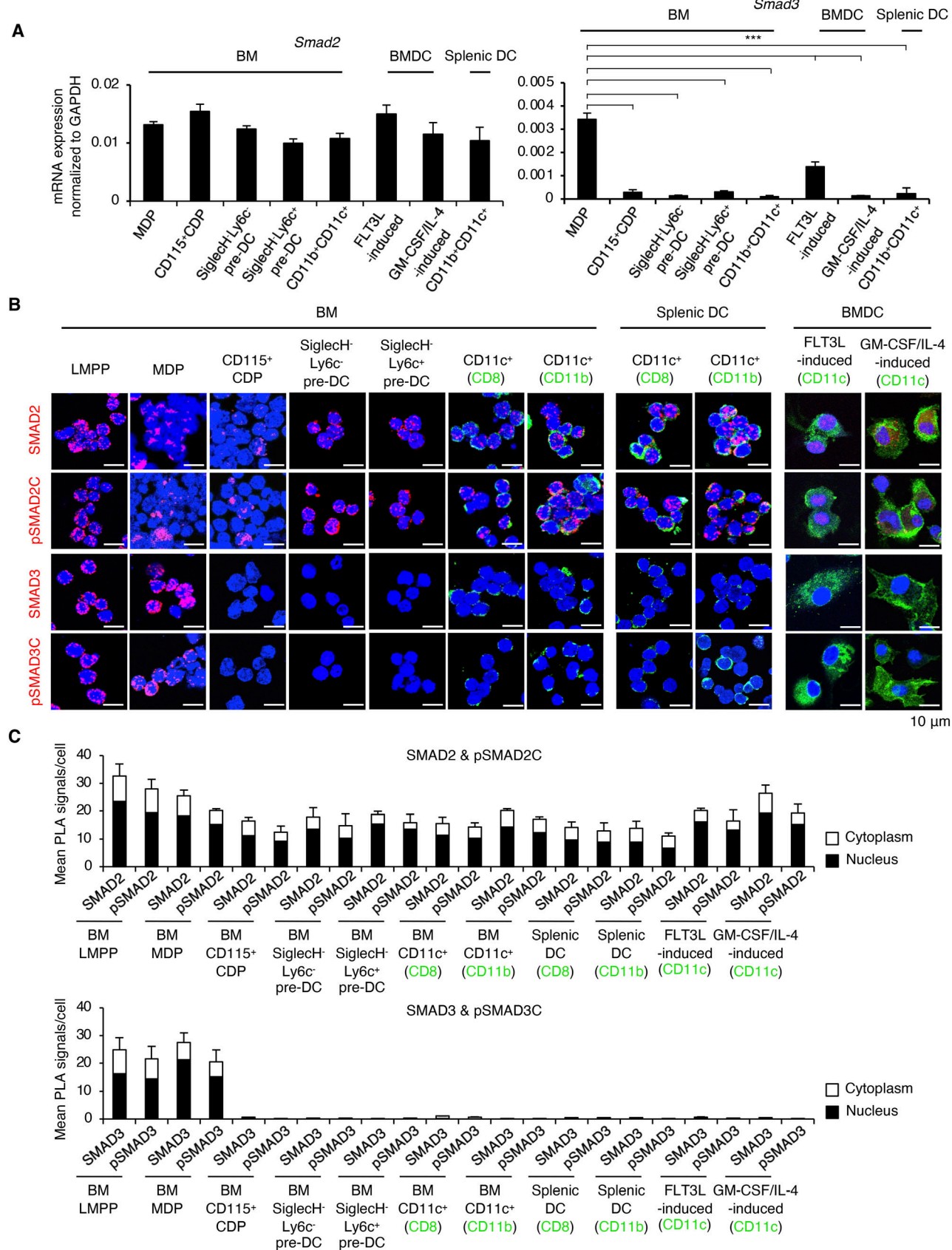

with a cDC potential (Schlitzer et al, 2015) were significantly increased in the BM of Smad3$^{-/-}$ mice (Fig 2D). In support of our findings, ImmGen consortium data demonstrate down-regulation of SMAD3 mRNA in the BM CDP in addition to CD11b$^+$ and CD8$^+$ DCs (Fig S2A).

We confirmed that the proportions of the haematopoietic progenitor cells detected as c-Kit$^+$Lin$^-$Sca-1$^+$ (KLS) or CD34$^+$ KLS cells, LMPPs, common myeloid progenitor as Lin$^-$Sca-1$^-$CD16/32$^-$CD34$^+$CD117$^+$, and granulocyte–macrophage progenitor as Lin$^-$Sca-1$^-$CD16/32$^+$CD34$^+$CD117$^+$ were unaltered in the BM of 8-wk-old Smad3$^{-/-}$ mice compared with littermate control Smad3$^{+/+}$ mice bred in the specific pathogen-free environment before the onset of any signs of inflammation (Yang et al, 1999; Yoon et al, 2015) (Fig S3B).

Immunophenotyping of DC progenitor subsets of Smad3-deficient mice suggests that SMAD3 deficiency facilitates cDC differentiation at the developmental stage between the MDP and CD115$^+$ CDP in the steady state.

### SMAD3-mediated TGF-β signalling inhibits cDC differentiation

We examined the direct effects of SMAD3 on cDC differentiation using FLT3L-induced BMDCs and GM-CSF plus IL-4–induced BMDCs (Xu et al, 2007) transfected with either SMAD3 DNA or SMAD3 siRNA 4 h before culture. Expression levels of FLAG-tagged SMAD3 were confirmed by flow cytometry (Fig S4A) and immunocytochemistry using the PLA (Fig S4B). SMAD3 mRNAs in the transfected BMDCs were confirmed by quantitative RT–PCR (RT–qPCR; Fig S4C). Flow cytometry data showed that the forced overexpression of SMAD3 resulted in a significantly decreased CD115$^+$ CDP in both FLT3L-induced or GM-CSF plus IL-4–induced BMDCs (Fig 3A, upper contour plots, and Fig S4D). In contrast, knockdown of SMAD3 resulted in a significantly increased CD115$^+$ CDP in both FLT3L-induced or GM-CSF plus IL-4–induced BMDCs (Fig 3A, lower contour plots, and Fig S4D) in consistent with the findings in Smad3-deficient mice (Fig 2B). The overexpression of SMAD3 reduced MHCII$^+$CD11c$^+$, CD11b$^+$CD11c$^+$, and CD24$^+$CD11c$^+$ cells (Naik et al, 2005) (Fig 3B, upper contour plots, and Fig S4F), whereas knockdown of SMAD3 increased MHCII$^+$CD11c$^+$, CD11b$^+$CD11c$^+$, and CD24$^+$CD11c$^+$ cells (Fig 3B, lower contour plots, and Fig S4F) with the gating procedures (Fig S4E). Transfection of pcDNA or control siRNA reduced the proportions of the cells highly positive for MHCII and CD11c; however, net percentages of MHCII$^+$CD11c$^+$ were not altered by transfection (Fig 3B and D).

SMAD3-knocked-down BMDCs developed significantly more dendrite formation, which is the most characteristic morphological feature of the cDC (Steinman & Cohn, 1973), whereas

SMAD3-overexpressed BMDCs did not develop dendrite formation (Fig 3C). SMAD2 and SMAD3 are the R-SMADs shared by TGF-β and activin among TGF-β superfamily cytokines (Batlle & Massague, 2019). Activin A (10 ng/ml) showed no effect on FLT3L or GM-CSF plus IL-4–induced BMDC differentiation (Fig 3D, lowest contour plots, and Fig S4G). A high concentration of TGF-β1 (5 ng/ml) completely blocked FLT3L or GM-CSF plus IL-4–induced Smad3$^{+/+}$ BMDC differentiation, which was abolished in Smad3$^{-/-}$ BMDCs (Fig 3D, middle contour plots, and Fig S4G), indicating that the potent inhibitory effect of high-dose TGF-β on DC differentiation is SMAD3-dependent. These data show that SMAD3-mediated TGF-β signalling inhibits cDC differentiation.

### SMAD3-mediated TGF-β signalling down-regulates cDC-related genes

To identify the target genes of SMAD3 to inhibit cDC differentiation, we screened essential cytokines, their signalling molecules, and the transcription factors for cDC differentiation: Flt3, Csf2ra, Pu.1, Gfi1, Irf2, Irf4, Irf8, Id2, Batf3, and RelB (Zhang et al, 2023), as well as TGF-β1, TGF-β2, and TGF-β3 in FLT3L-induced BMDCs and GM-CSF plus IL-4–induced BMDCs transfected with either SMAD3 DNA (black circles) or control pcDNA (white circles) (Figs 4A and B and S5). CD11b$^+$CD11c$^+$ cells were sorted from FLT3L-induced and GM-CSF plus IL-4–induced BMDCs before cell lysis. We found that the overexpression of SMAD3 significantly suppressed the mRNA expression of Flt3, Id2, and Irf4 in FLT3L-induced BMDCs (Fig 4A) and suppressed the mRNA expression of Flt3 and Irf4 in GM-CSF plus IL-4–induced BMDCs (Fig 4B). SMAD3 did not affect the Id2 mRNA expression in GM-CSF plus IL-4–induced BMDCs (Fig S5), which are comprised of both cDCs and monocyte-derived macrophages (Helft et al, 2015).

We examined the requirement of SMAD3 for TGF-β1 to suppress these identified cDC-related genes: Flt3, Id2, and Irf4 (Takagi et al, 2011). TGF-β1 suppressed the mRNA expression of these cDC-related genes in a dose-dependent manner in both FLT3L-induced and GM-CSF plus IL-4–induced Smad3$^{+/+}$ BMDCs, which was abolished in Smad3$^{-/-}$ BMDCs (Fig 4C and D).

To confirm the effect of SMAD3 on the expression of the identified cDC-related genes in vivo, we compared the mRNA expression levels of Flt3, Id2, and Irf4 in cDCs and their progenitor cells in the BM and spleens of Smad3$^{-/-}$ mice (black bars) and the littermate control Smad3$^{+/+}$ mice (white bars) (Fig 4E). Smad3$^{-/-}$ CD11b$^+$ cDCs and SiglecH$^-$Ly6C$^+$ pre-cDCs expressed significantly higher levels of Flt3, Id2, and Irf4. Smad3$^{-/-}$ CD115$^+$ CDPs expressed

**Figure 1. SMAD3 is selectively down-regulated in cDCs, SiglecH$^-$ pre-DCs, and CD115$^+$ CDPs.** Lin$^-$Sca-1$^+$CD34$^+$CD117$^+$CD135$^+$ (LMPP), Lin$^-$CD117$^{hi}$CD135$^+$CD115$^+$Sca-1$^-$ (MDP), Lin$^-$CD117$^{int}$CD135$^+$CD115$^+$CD127$^-$ (CD115$^+$ CDPs), SiglecH$^-$Ly6C$^-$/SiglecH$^-$Ly6C$^+$ CD11c$^+$MHCII$^-$CD135$^+$CD172α$^-$ (pre-DCs), BM CD8$^+$CD11c$^+$, CD11b$^+$CD11c$^+$ cDCs, FLT3L-induced BMDCs, GM-CSF plus IL-4–induced BMDCs, splenic CD8$^+$CD11c$^+$, and CD11b$^+$CD11c$^+$ cDCs were sorted using the MACS system and FACSAria III. **(A)** Expression of SMAD2 and SMAD3 mRNAs was measured by RT–qPCR. Data are the average of the triplicates of one representative out of five independent experiments. **(B)** Protein expression of SMAD3, C-terminally phosphorylated SMAD3 (pSMAD3C), SMAD2, and C-terminally phosphorylated SMAD2 (pSMAD2C) was measured by the proximity ligation assay. The nucleus was stained with DAPI. CD8, CD11b, and CD11c were stained with Alexa Fluor 488 (green). Scale bars represent 10 μm. Data are the representative images of five independent experiments. **(C)** Graphs show the quantification of SMAD3, pSMAD3C, SMAD2, and pSMAD2C determined by the fluorescence intensities of the PLA. Red dots in the nucleus (black) and cytoplasm (white) in 10 fields from one representative experiment out of five independent experiments were quantified. Graphs show means + s.d. P-values were calculated by a two-tailed unpaired t test for (A). ***P < 0.0005.

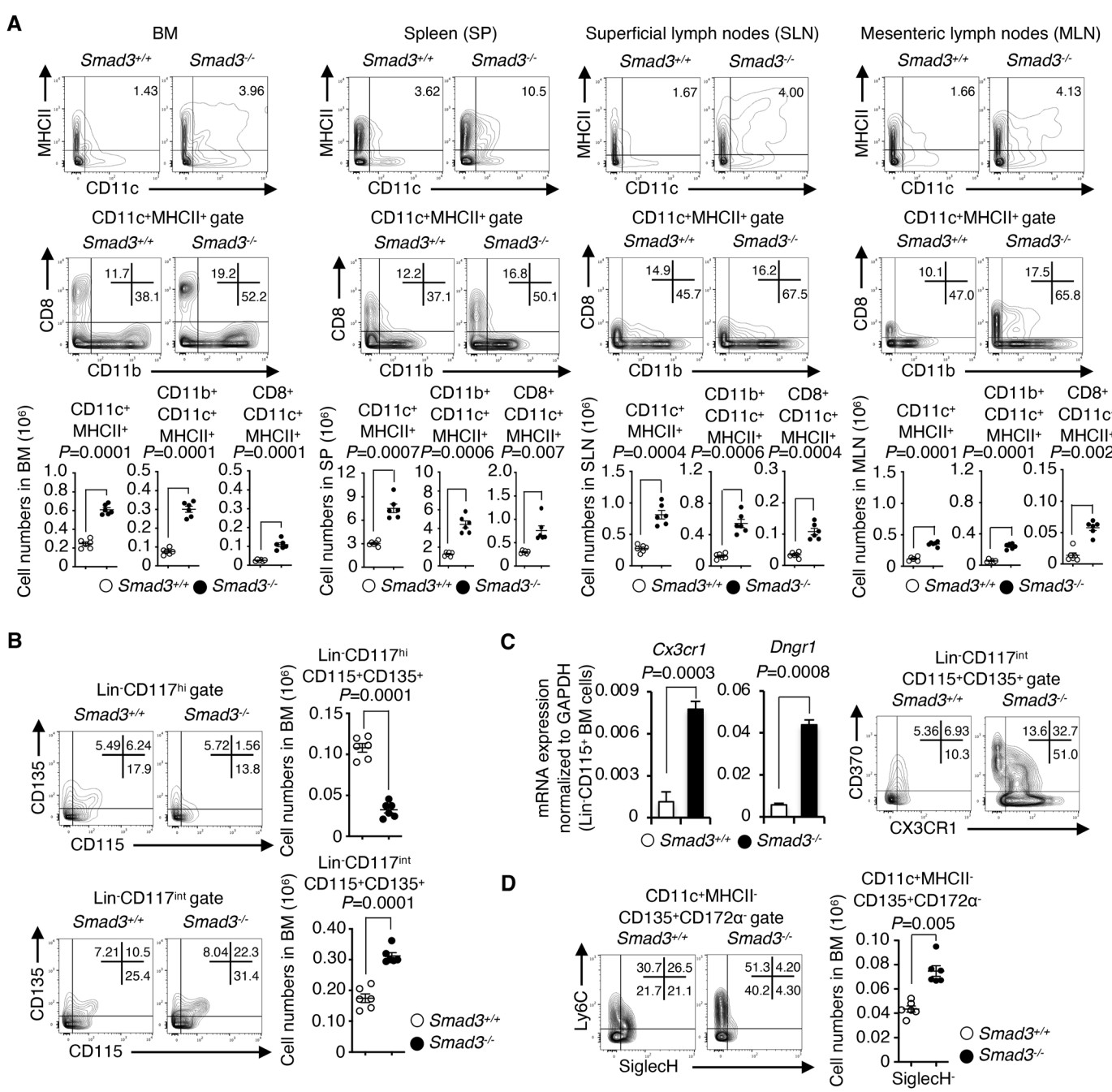

**Figure 2. cDCs and cDC progenitors: CD115+ CDP (Lin−CD117int), SiglecH−Ly6C+, and SiglecH−Ly6C− pre-DCs are increased in Smad3−/− mice.**
Immunophenotyping of *Smad3−/−* (black circles) or *Smad3+/+* (white circles) mice (n = 6/genotype) was performed using flow cytometry. **(A)** Representative contour plots show CD11c+MHCII+, CD11b+CD11c+MHCII+, and CD8+CD11c+MHCII+ cells in BM, spleens (SP), superficial lymph nodes, and mesenteric lymph nodes. Dots in the graphs show the numbers of CD11c+MHCII+, CD11b+CD11c+MHCII+, and CD8+CD11c+MHCII+ cells in BM, SP, superficial lymph nodes, and mesenteric lymph nodes of each mouse. Horizontal bars show means. **(B)** Representative contour plots show the expression of CD115/CD135 in Lin−Sca-1−CD117hi or Lin−Sca-1−CD117int gates. Dots in the graphs show the numbers of Lin−Sca-1−CD117hiCD115+CD135+ MDPs and Lin−Sca-1−CD117intCD115+CD135+ CDPs in the BM of each mouse. **(C)** Bar graphs show the expression levels of *Cx3cr1* and *Dngr1* mRNA in Lin−CD115+ BM cells detected using RT–qPCR with means + s.d. Black bars represent *Smad3−/−*, whereas white bars represent *Smad3+/+* mice. Representative contour plots show CX3CR1+CD370+Lin−CD117intCD115+CD135+ BM cells. **(D)** Representative contour plots show the expression of SiglecH/Ly6C in the CD11c+MHCII−CD135+CD172α− pre-DC gate. Dots in the graphs show the numbers of CD11c+MHCII−CD135+CD172α−SiglecH− cells in BM. *P*-values were calculated by a two-tailed unpaired *t* test.

higher levels of *Id2* and *Irf4*. *Smad3−/−* SiglecH−Ly6C− pre-DCs and CD8+ cDCs expressed higher levels of *Flt3* and *Id2* compared with the *Smad3+/+* littermate control (Fig 4E).

These data indicate that SMAD3-mediated TGF-β signalling inhibits cDC differentiation by repressing cDC-related genes such as *Flt3*, *Id2*, and *Irf4*.

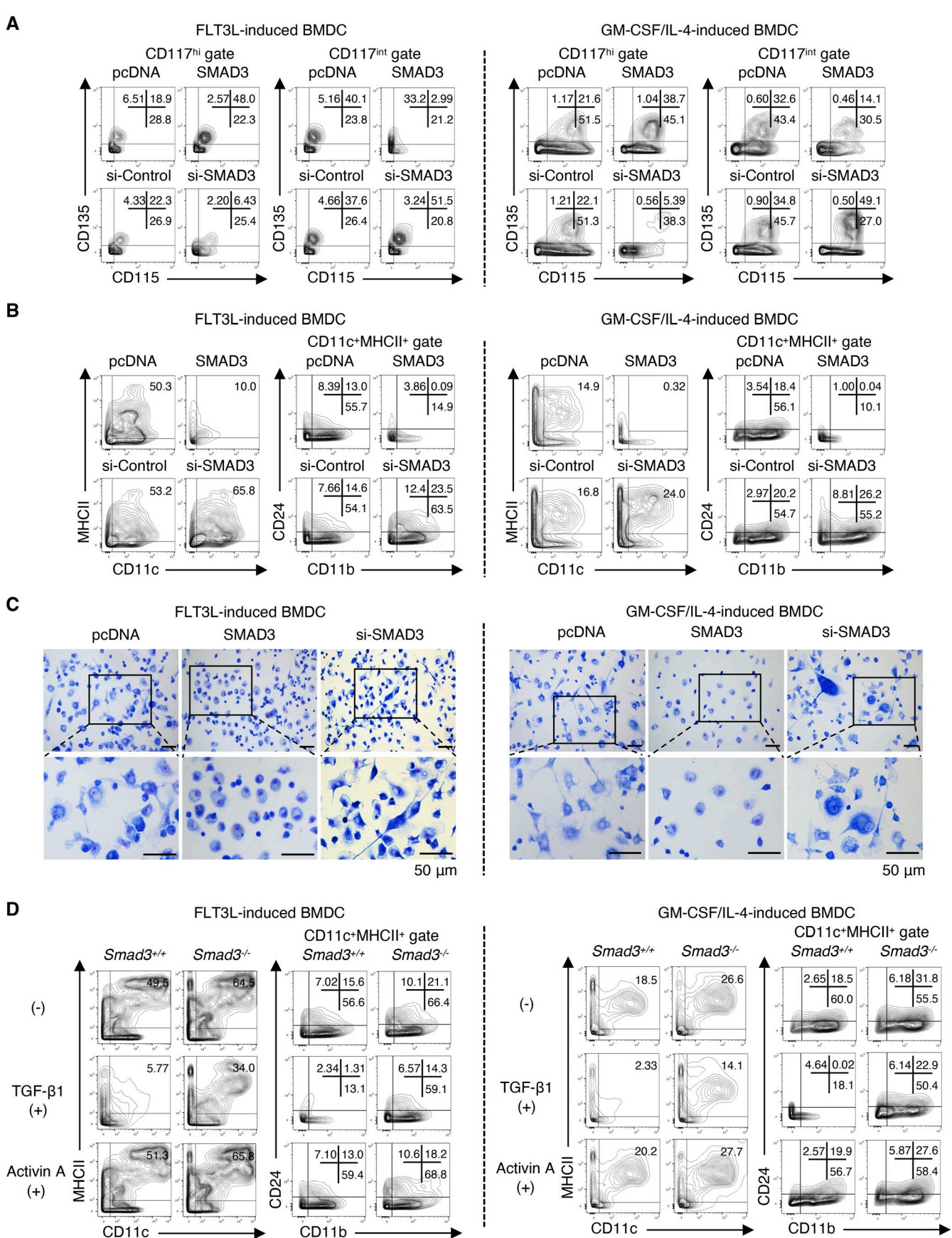

## STAT3 and c-SKI repress the transcription of SMAD3 for cDC differentiation

We next investigated the mechanisms whereby SMAD3 is down-regulated for cDC differentiation. The ligand of CD115, M-CSF, in-duces STAT3 activation in macrophages (Novak et al, 1995). STAT3 as the signalling molecule of FLT3L and GM-CSF (Onai et al, 2006; Li & Watowich, 2013; Wan et al, 2013) is essential for FLT3L-responsive DC progenitor proliferation (Laouar et al, 2003). Therefore, we exam-ined the effect of STAT3 on SMAD3 expression in BMDCs transfected with STAT3 or control pcDNA and siSTAT3 or control siRNA. Ex-pression levels of STAT3 mRNA in the transfected BMDCs were confirmed by RT–qPCR (Fig S6A). The overexpression of STAT3 suppressed, whereas knockdown of STAT3 up-regulated the ex-pression of *Smad3* mRNA in both FLT3L-induced and GM-CSF plus IL-4–induced BMDCs (Fig 5A).

We examined whether and how STAT3 regulates the *Smad3* gene promoter activity using the *Smad3* gene promoter luciferase re-porter construct spanning 2 kb upstream of the first exons of the *Smad3* gene transfected in FLT3L-induced or GM-CSF plus IL-4–induced BMDCs. SMAD2, SMAD3, and SMAD4 synergistically induced the *Smad3* promoter activity (Fig 5B, white bars), which was sup-pressed by STAT3 in both FLT3L-induced or GM-CSF plus IL-4–induced BMDCs (Fig 5B, light grey bars). To identify a corepressor of STAT3, we screened the representative transcriptional repressors of R-SMADs: SKI/SnoN and TGIF (Deheuninck & Luo, 2009; Heldin & Moustakas, 2012; Batlle & Massague, 2019). Among them, c-SKI showed the synergy with STAT3 to repress the *Smad3* promoter activity (Fig S6B). The overexpression of c-SKI exerted the repressive effect on the *Smad3* promoter activity, which was strengthened in synergy with STAT3 (Fig 5B, thick grey and black bars). Knockdown of c-SKI by siRNA completely abolished the repressive effect of STAT3 on the SMAD2/3/4-induced *Smad3* promoter activation (Fig 5C, black bars). In contrast, c-SKI alone retained the repressive effect on the SMAD2/3-induced *Smad3* promoter activation when STAT3 was knocked down, although the repressive effects of c-SKI and STAT3 were more effective in synergy (Fig 5D).

We performed chromatin immunoprecipitation followed by se-quencing (ChIP-seq) with CD11b[+] FLT3L-induced BMDCs to identify the enriched loci for DNA-binding STAT3 and c-SKI in association with the histone modification status within the *Smad3* coding and flanking regions (Fig 5E). ChIP-seq showed that the whole *Smad3* coding and flanking regions were epigenetically inactive, being highly enriched in trimethylated histone H3 lysine 27 (H3K27me3), which is associated with repression by polycomb group complexes. Transcriptionally active epigenetic marks such as acetylated lysine 23 (H3K23Ac) and trimethylated histone H3 lysine 4 (H3K4me3) were minor, but there were some bivalent sites enriched in both H3K27me3 and H3K4me3. Bivalent chromatin modification state balances the expression or repression of the important regulatory genes during cell differentiation (Macrae et al, 2023). We found that STAT3 and c-SKI bound to the bivalent sites within the proximal region of the *Smad3* gene. We then analysed the de novo enriched motifs using HOMER and searched the JASPAR CORE database. A canonical DNA-binding motif of STAT3 is TTCnnnGAA (Seidel et al, 1995). A reported DNA-binding motif of c-SKI is GTCTAGAC in chicken embryo fibroblasts (Nicol & Stavnezer, 1998). A de novo motif analysis showed that the STAT3-binding motif contained half-sequences: TTCC (Hutchins et al, 2013) (Fig 5F), and the c-SKI–binding motif contained the GTCTAG element (Fig 5G) in CD11b[+] FLT3L-induced BMDCs.

ChIP–qPCR for the proximal promoter region of the *Smad3* gene validated the results of ChIP-seq, showing that STAT3 and c-SKI bound to the same sites with SMAD2 in CD11b[+] FLT3L-induced BMDCs (−1,196 to −1,003 and −220 to −28) (Fig 5H), which were epigenetically inactive with H3K27me3 (Fig 5I).

These data indicate that STAT3 in synergy with c-SKI represses canonical SMAD pathway–induced transcription of the *Smad3* gene for cDC differentiation.

## c-SKI is required for STAT3 to interact with SMAD2 in cDCs

As a consequence of repression of SMAD3 by STAT3 and c-SKI, SMAD2 is the remained R-SMAD in cDC precursors and cDCs. We sought to confirm the physiological interactions among STAT3, c-SKI, and SMAD2 in MDPs and cDC precursors sorted from BM, cDCs sorted from BM and spleens, and FLT3L-induced or GM-CSF plus IL-4–induced BMDCs using the PLA. The PLA showed the close prox-imity between c-SKI and STAT3 and the close proximity between c-SKI and SMAD2 in CD115[+] CDPs, SiglecH[−] pre-DCs, CD11c[hi] BMDCs, and CD11b[+]CD11c[+] splenic cDCs, which was not observed in MDPs (Fig 6A). Knockdown of c-SKI by siRNA abolished the interaction between SMAD2 and STAT3 (Fig 6B), whereas knockdown of STAT3 had no effect on the interaction between c-SKI and SMAD2 in CD11c[hi] BMDCs (Fig 6C). We found that SMAD2 was C-terminally phosphorylated, indicating the presence of TGF-$\beta$ ligand–bound activated type I receptor signalling. Taken together, c-SKI is re-quired for STAT3 to interact with C-terminally phosphorylated SMAD2 in cDCs.

## Interaction of phosphorylated STAT3 with c-SKI is essential for repression of SMAD3 in cDCs

SMAD2 and SMAD3 interact with the N-terminal region of SKI, whereas SMAD4 interacts with the SAND-like domain of SKI, which blocks the ability of the SMAD complexes to activate transcription of TGF-$\beta$ target genes (Akiyoshi et al, 1999; Suzuki et al, 2004; Takeda et al, 2004; Batlle & Massague, 2019). STAT3 is phosphorylated at the C-terminal tyrosine residue (Y705) and serine residue (S727) upon stimulation with cytokines, protein tyrosine kinase receptors, or

---

**Figure 3. SMAD3 inhibits cDC differentiation.**
FLT3L-induced or GM-CSF plus IL-4–induced BMDCs were transfected with SMAD3 DNA or control pcDNA, and SMAD3 siRNA or control siRNA 4 h before culture and analysed on days 7–8. **(A)** Contour plots show the expression of CD115/CD135 in Lin[−]Sca-1[−]CD117[hi] or Lin[−]Sca-1[−]CD117[int] cells. **(B)** Contour plots show the expression of CD11c/MHCII and CD11b/CD24 in the CD11c[+]MHCII[+] gate. **(C)** May–Grunwald/Giemsa–stained BMDCs transfected with SMAD3 DNA, SMAD3 siRNA, or control pcDNA. Scale bars represent 50 $\mu$m. **(D)** Contour plots show the expression of CD11c/MHCII and CD11b/CD24 in the CD11c[+]MHCII[+] gate of *Smad3*[+/+] or *Smad3*[−/−] BMDCs treated with or without TGF-$\beta$1 (5 ng/ml) or activin A (10 ng/ml). Data are representative of five independent experiments.

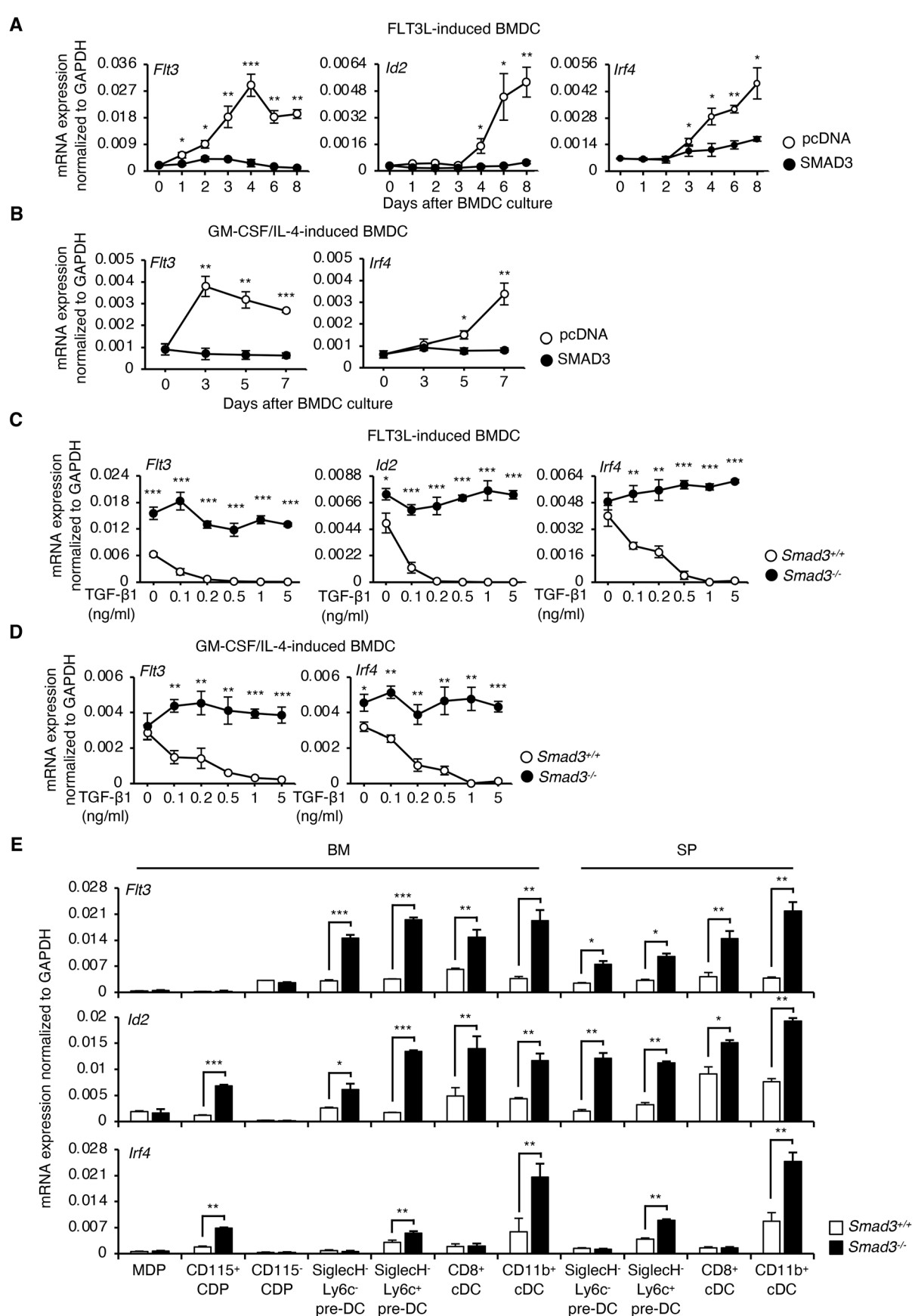

intracellular protein tyrosine kinases (Hillmer et al, 2016). Therefore, we investigated the mechanisms whereby c-SKI and STAT3 repress transcription of the *Smad3* gene using the *Smad3* gene promoter luciferase reporter assay with various combinations of mutants of c-SKI and STAT3 transfected in CD11b+ FLT3L-induced or GM-CSF plus IL-4–induced BMDCs.

A mutant of c-SKI that does not interact with SMAD2/3 (Δ2/3) failed to repress the *Smad3* promoter activity, whereas a mutant of c-SKI that does not interact with SMAD4 (W274E) (Wu et al, 2002; Nagata et al, 2006) retained the repressive effect on the *Smad3* promoter activity (Fig 7A). Inactive mutants of STAT3 at Y705 and S727 residues, Y705F and S727A, respectively, abolished the repressive effect on SMAD2/3-induced *Smad3* promoter activation in both CD11b+ FLT3L-induced and GM-CSF plus IL-4–induced BMDCs (Fig 7B).

We next examined how binding capacity of c-SKI with SMADs and phosphorylation status of STAT3 affect cDC differentiation using CD11b+ FLT3L-induced or GM-CSF plus IL-4–induced BMDCs transfected with various combinations of mutants of c-SKI and STAT3. Expression levels of FLAG-tagged c-SKI and mutants were confirmed by flow cytometry (Fig S6C) and immunocytochemistry using the PLA (Fig S6D). The overexpression of full-length c-SKI enhanced, whereas knockdown of c-Ski by siRNA suppressed, cDC differentiation (Fig 7C). Transfection of Δ2/3 failed to induce cDC differentiation, whereas transfection of W274E enhanced cDC differentiation, as well as full-length c-SKI (Fig 7C). In the same manner as c-Ski, the overexpression of STAT3 enhanced, whereas knockdown of STAT3 by siRNA suppressed cDC differentiation (Fig 7D). The enhancing effect of STAT3 on cDC differentiation was abolished by mutations of Y705F and S727A.

These data demonstrate that phosphorylated STAT3 at Y705 and S727 interacts with c-SKI and that SMAD2, but not SMAD4, is required to repress transcription of the *Smad3* gene for cDC differentiation.

## Discussion

Comprehensive comparative analyses using multi-omics techniques have deciphered complex transcriptional networks for DC differentiation and identified novel DC progenitors and subpopulations (Dress et al, 2018, 2019; Anderson et al, 2021; Cabeza-Cabrerizo et al, 2021; Ginhoux et al, 2022). Despite the ever-evolving diversity and complexity of cDC ontogeny, the CD115+ CDP has been defined as the early progenitor cells harbouring a potential for cDC differentiation (Onai et al, 2007; Nutt & Chopin, 2020). In this work,

we show that repression of the *Smad3* gene by STAT3 and c-Ski in the CD115+ CDP is essential for cDC differentiation.

TGF-β has been reported to play important regulatory roles in DC differentiation (Seeger et al, 2015). However, signalling mechanisms whereby TGF-β regulates the differentiation of DC subsets in the steady-state condition in vivo remained largely unknown. This study has identified SMAD3 as the repressor of cDC differentiation. TGF-β exerts the bidirectional effects on proliferation and differentiation versus quiescence depending on the haematopoietic stem cell (HSC) subtypes (Blank & Karlsson, 2015). The extracellular matrix stores and activates latent TGF-β in BM to transduce SMAD-mediated TGF-β signalling in HSCs and various haematopoietic progenitor cell populations (Söderberg et al, 2009; Massague & Xi, 2012; Robertson & Rifkin, 2016). TGF-β induces HSC hibernation (Yamazaki et al, 2009). TGF-β-SMAD3 signalling has been implicated to cooperate with FOXO signalling to quiescence and self-renewal of HSCs (Naka & Hirao, 2017). Considering the crucial role of SMAD3 in maintaining stem cell quiescence reported in these previous studies, nuclear localization of R-SMADs in freshly isolated BM progenitor cells (Fig 1B) suggests that the SMAD-mediated canonical TGF-β pathway maintains homeostasis of the early DC progenitors upstream of MDPs.

TGF-β receptor–regulated SMADs: SMAD2 and SMAD3, are ubiquitous and constitutive in normal cells in general, whereas their loss is frequently observed in various cancers. They have high amino acid sequence identity in their MH2 domains containing two C-terminal serine residues, 465/467 (SMAD2) and 423/425 (SMAD3); nevertheless, they regulate the same or distinct sets of TGF-β target genes to exert redundant or distinct functions depending on the context (Brown et al, 2007; Heldin & Moustakas, 2012; Batlle & Massague, 2019; Miyazawa et al, 2024). Precise mechanisms of how they are selected for differential functions by the context are largely undetermined. Down-regulation of SMAD3 expression by TGF-β through decreased transcription and/or increased ubiquitination has been reported in some normal epithelial cells such as lung epithelial cells (Yanagisawa et al, 1998) and human glomerular mesangial cells during epithelial-to-mesenchymal transition (Poncelet et al, 2007). This study is the first to show the transcriptional repression of the *Smad3* gene in haematopoietic cell lineages to induce normal development of cDC subsets in the steady state. These data suggest that transcriptional repression of SMAD3 is the main mechanism to select the optimal R-SMAD in response to TGF-β by the context.

We have clarified the mechanism of how SMAD3 is down-regulated for cDC differentiation. We found that STAT3 transcriptionally repressed SMAD3 to derepress cDC-related genes: FLT3, ID2,

**Figure 4. SMAD3-mediated TGF-β signalling down-regulates cDC-related genes.**
**(A)** Expression levels of *Flt3*, *Id2*, and *Irf4* mRNA in FLT3L-induced BMDCs transfected with SMAD3 DNA or control pcDNA 4 h before culture and analysed on days 1, 2, 4, 6, and 8. **(B)** Expression levels of *Flt3* and *Irf4* mRNA in GM-CSF plus IL-4–induced BMDCs transfected with SMAD3 DNA or control pcDNA 4 h before culture and analysed on days 3, 5, and 7. Black circles represent SMAD3 DNA, whereas white circles represent pcDNA. **(C)** Expression levels of *Flt3*, *Id2*, and *Irf4* mRNA in (C) FLT3L-induced BMDCs or (D) GM-CSF plus IL-4–induced BMDCs generated using *Smad3−/−* (black circles) or *Smad3+/+* (white circles) BM treated with the indicated concentrations of TGF-β1. **(A, B, C, D)** CD11b+CD11c+ cells were sorted from FLT3L-induced BMDCs and GM-CSF plus IL-4–induced BMDCs before cell lysis. **(E)** Expression levels of *Flt3*, *Id2*, and *Irf4* mRNA in MDPs, CD115+ CDPs, CD115− CDPs, SiglecH−Ly6C− pre-DCs, SiglecH−Ly6C+ pre-DCs, CD8+ cDCs, CD11b+ cDCs from BM, SiglecH−Ly6C− pre-DCs, SiglecH−Ly6C+ pre-DCs, and CD8+ and CD11b+ cDCs from spleens of *Smad3−/−* (black bars) or *Smad3+/+* (white bars) mice. Expression levels of mRNA were determined by RT–qPCR. Data are representative of three independent experiments in triplicate. Graphs show means + or ± s.d. *P*-values were calculated by a two-tailed unpaired *t* test. *P < 0.05, **P < 0.01, and ***P < 0.0005.

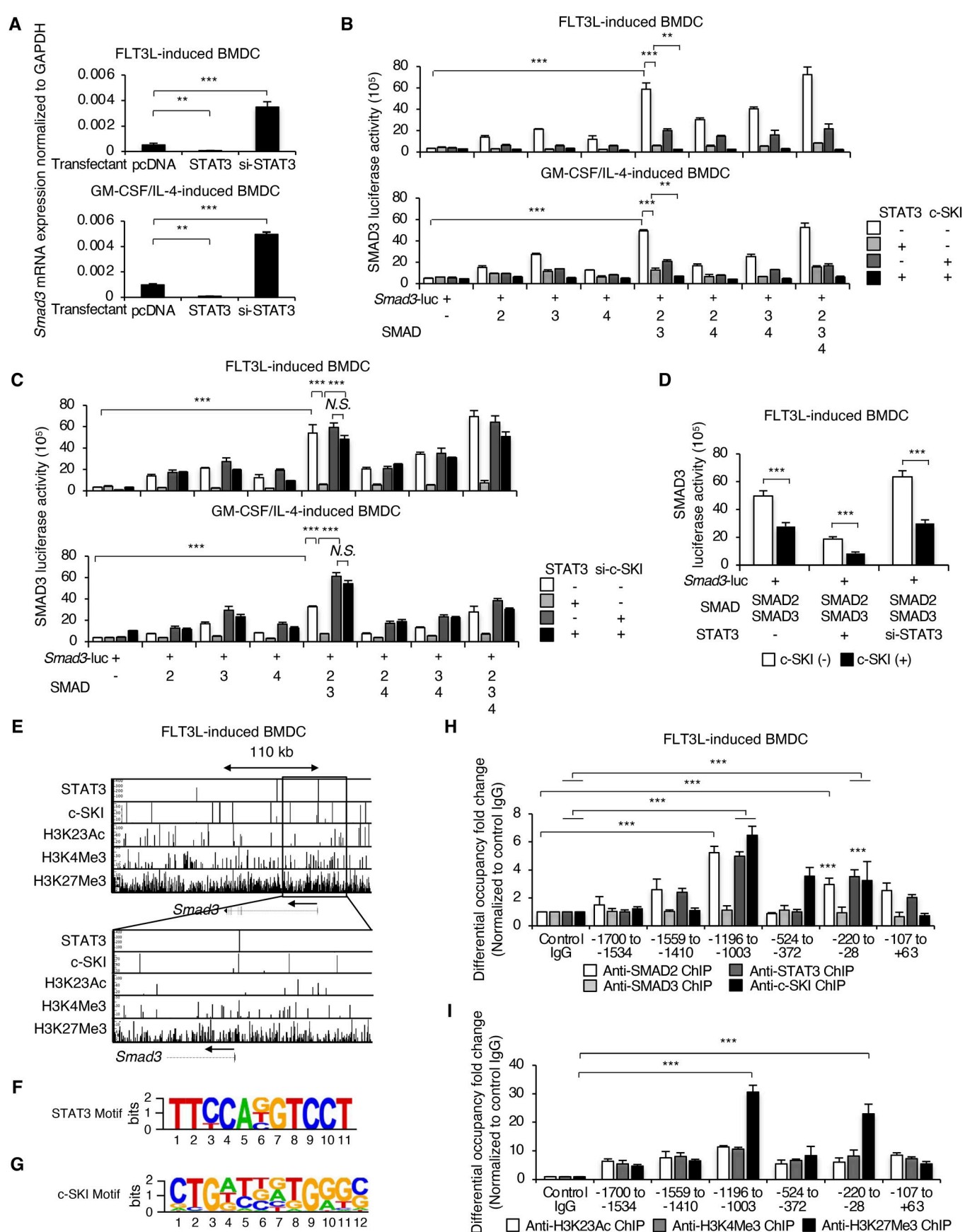

and IRF4 in cDCs, SiglecH⁻ pre-DCs, and CD115⁺ CDPs. The necessity of STAT3 in DC development has been established by the findings such as the loss of cDCs resulted from STAT3 deletion in vivo (Laouar et al, 2003) and promotion of DC maturation from the progenitors by STAT3 overexpression (Onai et al, 2006). Essential cytokines for cDC development induce phosphorylation of STAT3 in DC progenitor cells; engagement of FLT3L and FLT3 on DC precursors such as MDPs, CDPs, and pre-DCs (Onai et al, 2007; Liu et al, 2009) induces rapid phosphorylation of STAT3 resulting in FLT3L-responsive DC progenitor proliferation (Li & Watowich, 2013); STAT3 transiently activated by GM-CSF promotes differentiation of myeloid lineages including cDCs (Merad et al, 2013; Wan et al, 2013); and colony-stimulating factor, the ligand of CD115, induces STAT3 activation (Novak et al, 1995). This study has revealed the mechanism of how phosphorylated STAT3 at Y705 and S727 residues induces cDC differentiation through repressing transcription of SMAD3.

We further show that c-SKI is required for STAT3 to repress SMAD3 for cDC differentiation. SKI and the closely related SnoN oncogenes act as transcriptional corepressors in TGF-β signalling through interaction with SMADs (Akiyoshi et al, 1999; Wu et al, 2002; Suzuki et al, 2004; Nagata et al, 2006; Tecalco-Cruz et al, 2018). Although SKI is more widely expressed than SnoN in mature haematopoietic cells and plays crucial roles in haematopoiesis and myeloproliferative diseases (Pearson-White et al, 1995; Singbrant et al, 2014), its roles in differentiation and functions of immune cells remained largely undetermined. We have previously shown that SKI and SnoN oncoproteins cooperate with phosphorylated STAT3 in an adenocarcinoma lung cancer cell line, HCC827, to repress transcription of the *Smad3* gene, which renders the sensitive cells resistant to gefitinib (Makino et al, 2017). Here, we show that c-SKI, but not SnoN, is indispensable for STAT3 to repress the *Smad3* gene for cDC differentiation rather than playing a conventional role as a transcriptional corepressor of SMADs. In haematopoietic cells, SKI represses retinoic acid receptor signalling (Dahl et al, 1998), which enhances SMAD3/SMAD4-driven transactivation (Pendaries et al, 2003). SKI induces a gene signature associated with HSCs and myeloid differentiation, as well as hepatocyte growth factor signalling (Singbrant et al, 2014). These previous reports and our finding suggest the possibility that hepatocyte growth factor signals via STAT3 (Schaper et al, 1997) might induce synergy with c-SKI to repress SMAD3 towards myeloid differentiation. Distinctions in the binding sites of SMADs, STAT3, and c-SKI in the *Smad3* promoter regions in cDCs and the HCC827 lung cancer cell line are consistent with the previous report showing that cell type–specific master transcription factors direct SMAD3 to distinct specific

binding sites to determine cell type–specific responses to TGF-β signalling (Mullen et al, 2011).

TGF-β had been reported to direct cDC differentiation from the CDP by inducing the essential factors for cDC differentiation such as IRF4, IRF8, RelB, ID2, and FLT3 (Felker et al, 2010; Sere et al, 2012). The seeming discrepancy between these reports and this study may be attributed to two possibilities. We observed that SMAD2 in close proximity with STAT3 and c-Ski is C-terminally phosphorylated (Fig 6), which indicates the presence of TGF-β receptor signalling. Future studies are required to explore the roles of SMAD3-independent TGF-β signalling in cDC differentiation. The other possibility is attributed to their two-step amplification and differentiation in vitro culture systems. SCF, high-dose IL-6, and insulin-like growth factor-1 contained in the first-step amplification culture might have enriched the CD115⁺ CDP. The second-step differentiation culture contains GM-CSF and IL-4, which induce BMDCs with cDC features. They reported that ID2 was up-regulated in TGF-β–treated CDPs (Hacker et al, 2003; Felker et al, 2010). Neither TGF-β nor SMAD3 affected the expression of ID2 in GM-CSF plus IL-4–induced BMDCs without first-step amplification culture (Fig S5). We have observed that TGF-β rather repressed ID2 in FLT3L-induced BMDCs (Fig 4A and C), which is consistent with the previous report on epithelial cells (Zavadil & Bottinger, 2005). The expression of ID2 was also significantly up-regulated in cDCs, SiglecH⁻ pre-DCs, and CD115⁺ CDPs of *Smad3⁻/⁻* mice in vivo (Fig 4E). These previous reports show the effect of TGF-β on already committed cDC precursors induced by their two-step culture protocol, whereas this study shows the regulatory effects of SMAD3-mediated TGF-β signalling on upstream DC progenitors.

In summary, we demonstrate the roles of SMAD3-mediated TGF-β signalling in murine cDC differentiation in the steady state. SMAD3 represses cDC-related genes so that repression of SMAD3 by phosphorylated STAT3 in cooperation with c-SKI is required for commitment to cDCs from CD115⁺ CDPs and SiglecH⁻ pre-DCs. The results of this study would provide the basis for future research on the roles of SMAD-mediated TGF-β signalling in differentiation and functions of effector cDC subsets in the inflamed and pathological settings.

## Materials and Methods

### Mice

Age-matched female *Smad3⁺/⁺, ⁻/⁻* mice (8 wk, six mice/genotype) were maintained and used for experiments according to the ethical

**Figure 5. STAT3 and c-SKI repress transcription of the Smad3 gene.**
**(A)** Expression levels of *Smad3* mRNA in CD11b⁺ FLT3L-induced or CD11b⁺ GM-CSF plus IL-4–induced BMDCs transfected with STAT3 DNA, control pcDNA, or STAT3 siRNA were determined by RT–qPCR. BMDCs were transfected with the *Smad3* promoter luciferase reporter construct with the indicated combinations of siRNA and DNA constructs 4 h before culture and analysed on day 7. CD11b⁺CD11c⁺ cells were sorted from FLT3L-induced BMDCs and GM-CSF plus IL-4–induced BMDCs before cell lysis. **(B, C, D)** The *Smad3* promoter luciferase reporter assays (triplicate, one representative of three independent experiments shown) were performed using the cells transfected with the indicated combinations of (B) SMAD2, SMAD3, SMAD4, STAT3, and c-SKI, (C) SMAD2, SMAD3, SMAD4, STAT3, and c-SKI siRNA, and (D) SMAD2, SMAD3, STAT3, siRNA, and c-SKI. **(E, H, I)** ChIP-seq and (H, I) ChIP-qPCR (triplicate, one representative of three independent experiments shown) using the antibodies against (E) STAT3 and c-SKI, (H) SMAD2, SMAD3, STAT3, and c-SKI, (E, I) H3K23Ac, H3K4me3, and H3K27me3 show the binding of SMAD2, SMAD3, STAT3, c-SKI, and the histone marks in the *Smad3* promoter region in CD11b⁺ FLT3L-induced BMDCs. **(F, G)** Motif enrichment analysis using HOMER show de novo motifs of (F) STAT3 and (G) c-SKI in the *Smad3* promoter in CD11b⁺ FLT3L-induced BMDCs. *P*-values were calculated by a two-tailed unpaired *t* test. \*\**P* < 0.01 and \*\*\**P* < 0.0005.

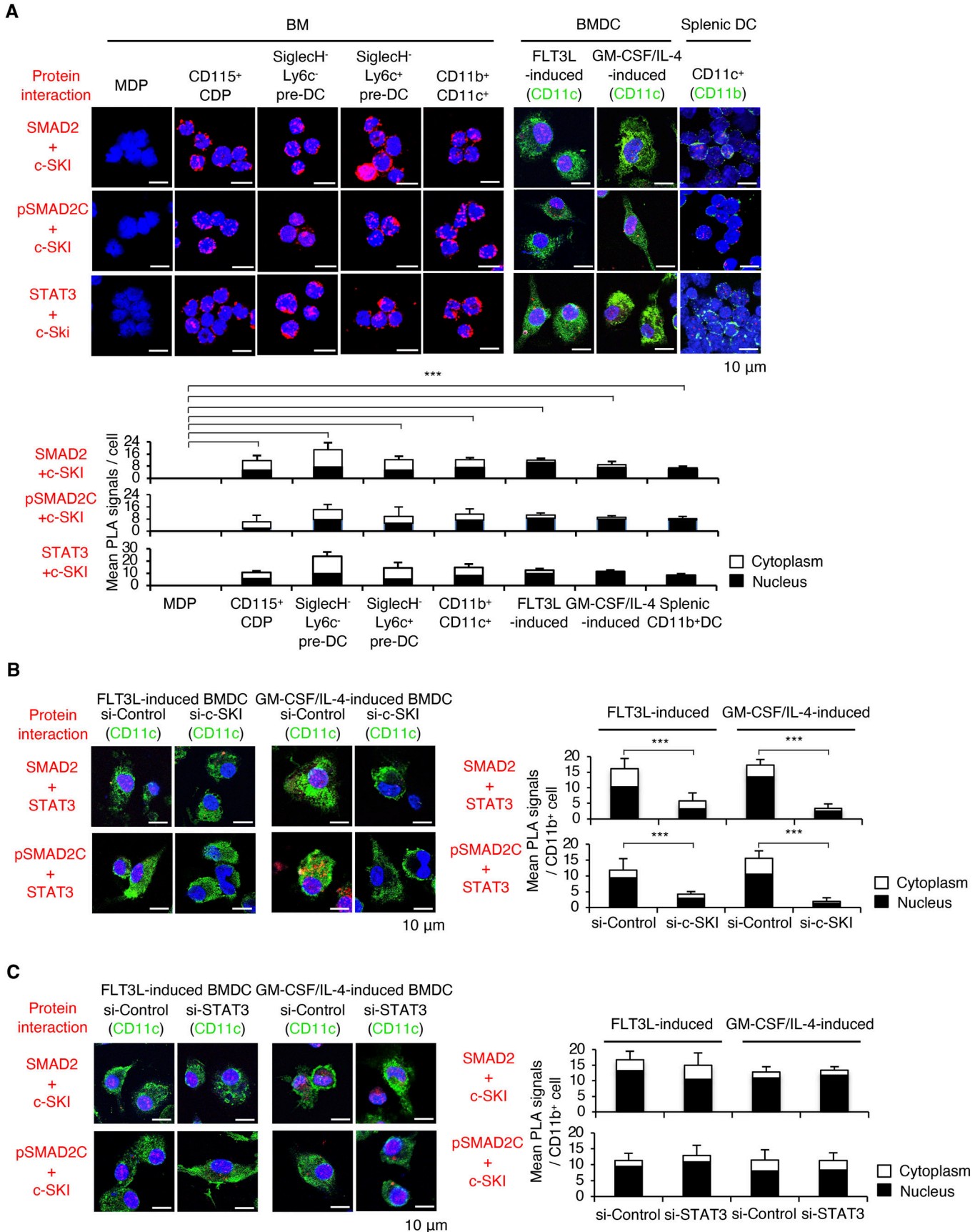

guidelines for animal experiments and the safety guidelines for gene manipulation experiments at the University of Tsukuba, Japan, Tokyo Medical University, Japan, and Konkuk University, Korea, under approved animal study protocols. No statistical method was used to predetermine sample sizes.

## Generation of mouse BMDCs

BM cells were flushed from femurs and tibias. BM cells counted using a Countess 3 automated cell counter (AMQAX2000; Thermo Fisher Scientific) were cultured in RPMI 1640 (SH30255.01; GE Healthcare), supplemented with 10% heat-inactivated FBS (12483-020; Gibco), 50 $\mu$M 2-mercaptoethanol (M314A; Sigma-Aldrich), 1 mM sodium pyruvate (S8636; Sigma-Aldrich), and 1% penicillin/streptomycin (SV30010; GE Healthcare), containing mouse GM-CSF (20 ng/ml; 315-03; PeproTech) plus IL-4 (20 ng/ml; 214-14; PeproTech) or FLT3L (200 ng/ml; 250-31L; PeproTech) at $1 \times 10^6$ cells/ml at 37°C with or without TGF-$\beta$1 (0.1, 0.2, 0.5, 1.0, and 5.0 ng/ml; 7666-MB; R&D Systems) or activin A (10 ng/ml; 120-14; PeproTech) for 7–8 d. For time course analyses (Fig 4A and B), BMDCs were cultured for 1–8 d.

## Flow cytometry

Mouse BMDCs, BM, spleens, and superficial and mesenteric lymph node cells were incubated on ice for 30 min with optimal concentrations of anti-mouse CD16/CD32 antibody (clone 2.4G2), biotin-conjugated mouse lineage panel (559142; BD Pharmingen), the fluorochrome-conjugated antibodies against anti-biotin streptavidin, anti-mouse CD34 (clone RAM34), anti-mouse CD127 (clone SB/199), anti-mouse CD117 (c-Kit) (clone 2B8), anti-mouse Ly-6A/E (Sca-1) (clone D7), anti-mouse CD115 (c-fms) (clone AFS98), anti-mouse CD135 (FLT3) (clone A2F10), anti-mouse CD370 (clone Clec9A), anti-mouse CX3CR1 (clone SA011F11), anti-mouse Ly6C (clone HK1.4), anti-mouse MHC Class II (clone I-A/I-E) (clone M5/1 114.15.2), anti-mouse CD11c (clone N418), anti-mouse CD8a (clone 53-6.7), anti-mouse CD24 (clone M1/69), anti-mouse CD11b (clone M1/70), anti-mouse CD172$\alpha$ (clone P84) and anti-mouse SiglecH (clone 551), and anti-tag FLAG (clone 8H8L17). All antibodies were purchased from BD Pharmingen, eBioscience, Invitrogen, or Bio-Legend. Dead cells were excluded by 7AAD (559925; BD Pharmingen). The samples were acquired by LSRFortessa (BD Bioscience) and analysed by FlowJo V10 (BD Bioscience). Gating procedures for flow cytometry data were performed according to the published protocol (Liu et al, 2020).

## Cell isolation

Spleens and LNs were chopped and digested with type III collagenase (LS004182; Worthington Biomedical Corporation) and DNase

I (11284932001; Roche) in RPMI 1640 containing 10% FBS for 30 min (spleens and LNs) at 37°C as described previously (Inaba et al, 2009). EDTA (5 mM; T4049; Sigma-Aldrich) was added for the final 5 min. Cell numbers were counted using a Countess 3 automated cell counter (Thermo Fisher Scientific). CD8$^+$CD11c$^+$ cells (130-091-169; Miltenyi Biotec), CD11b$^+$CD11c$^+$ cells (130-049-601, 130-108-338; Miltenyi Biotec), CD115$^+$ Lin$^-$ BM cells (130-096-354; Miltenyi Biotec), CD11b$^+$ cells, and CD4$^+$ T cells (130-049-601, 130-104-454; Miltenyi Biotec) were enriched using the MACS magnetic sorting system (Miltenyi Biotec). Purity was confirmed as >85%. To isolate LMPP, MDP, and CD115$^+$/CD115$^-$ CDP cells, Lin$^-$ cells were presorted from mouse BM using the mouse lineage cell depletion kit (130-090-858; Miltenyi Biotec) and MACS system (Miltenyi Biotec). Lin$^-$ cells were stained with fluorochrome-conjugated anti-mouse CD117 (clone 2B8), Sca-1 (clone D7), anti-mouse CD34 (clone RAM34), CD127 (clone SB/199), CD115 (clone AFS98), and CD135 (clone A2F10) antibodies for cell sorting using FACSAria III (BD Bioscience). To isolate pre-DCs, BM cells were stained with fluorochrome-conjugated anti-mouse CD11c (clone N418), MHC Class II (clone I-A/I-E) (clone M5/1 114.15.2), CD135 (clone A2F10), CD172$\alpha$ (clone P84), SiglecH (clone 551), and Ly6C (clone HK1.4) antibodies for cell sorting using FACSAria III. CD11b$^+$CD11c$^+$ cells were sorted from FLT3L-induced BMDCs and GM-CSF plus IL-4–induced BMDCs before cell lysis. All antibodies were purchased from BD Pharmingen, eBioscience, or BioLegend.

## Quantitative RT–PCR

The total RNA was extracted using TRIzol according to the manufacturer's instructions (15596-026; Invitrogen). The RNA was reverse-transcribed with a cDNA RT kit (18080200; Invitrogen). The mouse cDNA was quantitated by SYBR Green (4368708; Applied Biosystems) using ABI 7900 (Applied Biosystems). The following primers are listed in the table (Table S1). The relative mRNA levels to GAPDH were calculated by the comparative Ct method. Each experiment was performed in triplicate.

## Western blotting

Cells lysed in RIPA buffer were electrophoresed on 10% SDS–polyacrylamide gel, transferred to nitrocellulose membrane (Millipore), and probed with antibodies against anti-mouse SMAD2 (clone D43B4; Cell Signaling Technology), anti-mouse phospho-SMAD2 (S465/467) (clone E8F3R; Cell Signaling Technology), anti-mouse SMAD3 (clone EP568Y; Abcam), phospho-SMAD3 (S423/425) (clone EP823Y; Abcam), and $\beta$-actin (clone N-21; Santa Cruz Biotechnology). Blots were visualized using an ECL kit (GE Healthcare). Three independent experiments were performed.

---

**Figure 6.  Interaction of phosphorylated STAT3 with c-SKI and SMAD2 in cDCs.**
**(A, B, C)** Proximity between (A) SMAD2 and c-SKI, pSMAD2C and c-SKI, and STAT3 and c-SKI in MDPs, CD115$^+$ CDPs, SiglecH$^-$Ly6C$^-$ pre-DCs, SiglecH$^-$Ly6C$^+$ pre-DCs, CD11b$^+$ cDCs, FLT3L-induced BMDCs, GM-CSF plus IL-4–induced BMDCs, and splenic CD11b$^+$ cDCs; (B) STAT3 and SMAD2, STAT3 and pSMAD2C, (C) SMAD2 and c-SKI, and pSMAD2C and c-SKI in FLT3L-induced or GM-CSF plus IL-4–induced BMDCs transfected with the indicated c-SKI siRNA, and STAT3 siRNA or control siRNA was determined by the PLA (red). The nucleus was stained with DAPI. CD11c was stained with Alexa Fluor 488 (green). Red dots in the nucleus (black) and cytoplasm (white) in 10 fields were quantified. Scale bars represent 10 $\mu$m. Data are representative of three independent experiments. Graphs show means + s.d. $P$-values were calculated by a two-tailed unpaired $t$ test. ***$P$ < 0.0005.

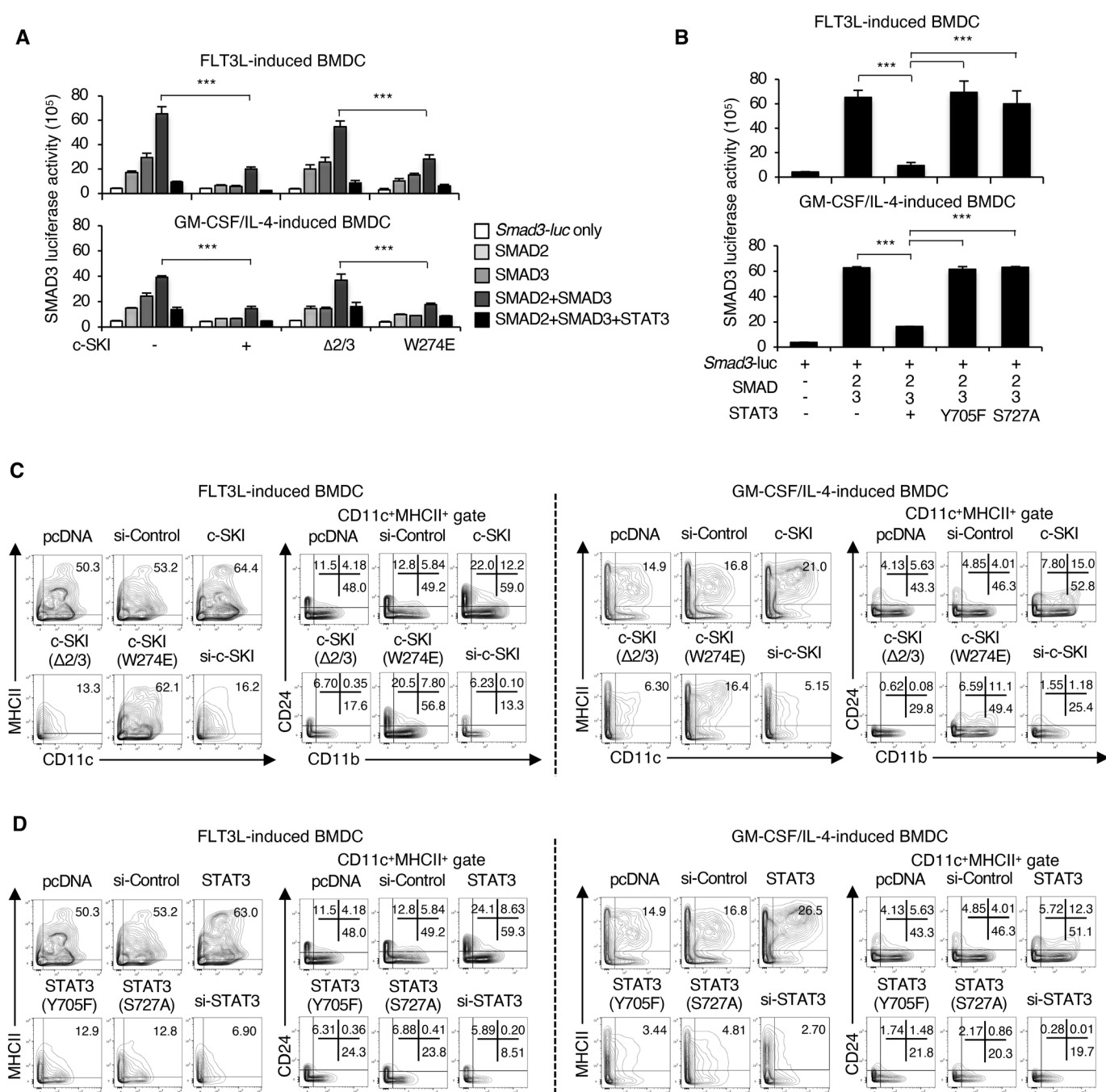

**Figure 7. Interaction of c-SKI with R-SMADs and phosphorylated STAT3 is essential for repression of SMAD3 and cDC differentiation.**
**(A)** Effects of c-SKI mutations (Δ2/3 that does not interact with Smad2/3 and W274E that does not interact with Smad4) on the *Smad3* promoter activity in CD11b⁺ FLT3L-induced or CD11b⁺ GM-CSF plus IL-4–induced BMDCs transfected with the indicated plasmids were determined by the *Smad3* promoter luciferase reporter assay. **(B)** Effects of STAT3 phosphorylation site-specific mutants (Y705F and S727A) on SMAD2/3-induced activation of the *Smad3* promoter constructs transfected with the indicated plasmids in CD11b⁺ FLT3L-induced or CD11b⁺ GM-CSF plus IL-4–induced BMDCs were determined by the *Smad3* promoter luciferase reporter assay. **(C)** Contour plots show the expression of CD11c/MHCII and CD11b/CD24 in the CD11c⁺MHCII⁺ gate of FLT3L-induced or GM-CSF plus IL-4–induced BMDCs transfected with the indicated c-SKI mutants, c-SKI siRNA and control pcDNA or control siRNA. **(D)** Contour plots show the expression of CD11c/MHCII and CD11b/CD24 in the CD11c⁺MHCII⁺ gate of FLT3L-induced or GM-CSF plus IL-4–induced BMDCs transfected with the indicated STAT3 phosphorylation site-specific mutants (Y705F and S727A) and STAT3 siRNA or control siRNA 4 h before culture and analysed on day 8. Luciferase reporter assays were performed in triplicate. Data are representative of three independent experiments. Graphs show means + s.d. *P*-values were calculated by a two-tailed unpaired *t* test. ***P < 0.0005.

## Immunocytochemistry

Lin$^-$CD117$^{hi}$CD135$^-$CD115$^-$CD127$^-$, Lin$^-$CD117$^+$Sca1$^+$CD34$^+$CD135$^+$, Lin$^-$CD117$^{hi}$CD135$^+$CD115$^+$CD127$^-$Sca-1$^-$, Lin$^-$CD117$^{int}$CD135$^+$CD115$^+$CD127$^-$, CD11c$^+$MHCII$^-$CD135$^+$ CD172$\alpha^-$SiglecH$^-$Ly6C$^-$, CD11c$^+$MHCII$^-$CD135$^+$CD172$\alpha^-$SiglecH$^-$Ly6C$^+$, CD8$^+$CD11c$^+$, CD11b$^+$CD11c$^+$ BM cells, CD8$^+$CD11c$^+$, CD11b$^+$CD11c$^+$ splenic cells, and FLT3L or GM-CSF plus IL-4–induced BMDCs were fixed with 4% PFA in phosphate-buffered saline for the May–Grunwald/Giemsa staining or PLA (Söderberg et al, 2006). For the PLA, fixed cells were permeabilized by 0.1% Triton X-100 in PBS, and stained with rabbit antibodies against SMAD2, SMAD3, phospho-SMAD2 (S465/467), phospho-SMAD3 (S423/425) (Cell Signaling Technology), STAT3 (clone C-20; Santa Cruz Biotechnology), c-SKI (clone 6D763; Santa Cruz Biotechnology), and FLAG (clone 8H8L17; Invitrogen). Subsequently, they were reacted with the Duolink In Situ PLA probe anti-rabbit PLUS (DUO92002; Sigma-Aldrich) and PLA probe anti-rabbit MINUS (DUO92005; Sigma-Aldrich), or PLA probe anti-mouse PLUS (DUO92001; Sigma-Aldrich) and PLA probe anti-mouse MINUS (DUO92004, Sigma-Aldrich), and signals were detected using In Situ Detection Reagents Red (DUO92008; Sigma-Aldrich) according to the manufacturer's fluorescence instructions. CD11c, CD11b, or CD8 cells were stained with Alexa Fluor 488 (green). The nucleus was stained with DAPI (blue). Slides were observed using a confocal microscope, LSM700 and LSM900, with a 374-nm axial resolution at 550 nm (Carl Zeiss). Signals in each field (n = 10 fields) were quantified using BlobFinder software V3.2 (Uppsala University). Five independent experiments were performed.

## Transfection

BM cells were transfected with 200 nM of a non-targeting siRNA (D-001810-01-05; Dharmacon), SMAD3 siRNA (L040706-00-0005; Dharmacon), STAT3 siRNA (L-040794-01-0005; Dharmacon), c-SKI siRNA (L-042265-01-0005; Dharmacon), empty pcDNA3 STAT3, STAT3 Y705F, S727A (submitted by J. Darnell; Addgene), FLAG-tagged SMAD2, FLAG-tagged SMAD3, FLAG-tagged SMAD4, FLAG-tagged c-SKI, FLAG-tagged c-SKI (Δ2/3 or W274E), FLAG-tagged TGIF, and FLAG-tagged SnoN in nucleofection buffer for 4 h before FLT3L-induced or GM-CSF plus IL-4–induced BMDC culture using Nucleofector I/II/2b (program Y-001), 4D-Nucleofector (program DK-100), and Amaxa Nucleofector kits for mouse dendritic cells (VPA-1011 and V4XP-4012; Lonza) with ~60–75% transfection efficiency according to the manufacturer's protocol. Knockdown efficiencies by siRNA were confirmed by RT–qPCR. Transfection efficiencies of FLAG-tagged plasmids were confirmed by detection of FLA using flow cytometry (Figs S4A and S6C) and immunocytochemistry by the PLA (Figs S4B and S6D), with ~55–60% and 75–80% of transfection efficiencies, respectively. Each experiment was performed in triplicate, and three independent experiments were performed.

## Luciferase assay

The 2,000-bp promoter region of SMAD3 was generated by PCR from genomic DNA using primers (Table S2). The product was verified by sequencing and was subcloned into a pGL3 basic firefly luciferase construct (E1751; Promega) using MluI and XhoI sites. The promoter constructs in various combinations with STAT3, STAT3 Y705F, S727A, FLAG-tagged SMADs, FLAG-tagged c-SKI, c-SKI (Δ2/3 or W274E), FLAG-tagged TGIF, FLAG-tagged SnoN, p300, empty pcDNA3, or non-targeting siRNA, STAT3 siRNA, c-SKI siRNA (Dharmacon RNA Technologies) plasmid were transfected in FLT3L-induced or GM-CSF plus IL-4–induced BMDCs with a control TK-pRL Renilla luciferase control reporter plasmid (E2231; Promega). Five to seven days after transfection, BMDCs were lysed by Dual-Luciferase Reporter Assay Kit (E1910; Promega), and the lysate was measured by a luminometer (Berthold Technologies). Each experiment was performed in triplicate.

## Chromatin immunoprecipitation

Chromatin was prepared from BMDCs. Immunoprecipitation was performed with antibodies against SMAD2 (Cell Signaling Technology), SMAD3 (Abcam), c-SKI, STAT3 (Santa Cruz Biotechnology), trimethyl histone H3K4 (clone C42D8), trimethyl histone H3K27 (C36B11; Cell Signaling Technology), and acetyl histone H3K23 (17-10112; Millipore) using Pierce Magnetic ChIP Kit (26157; Thermo Fisher Scientific) according to the manufacturer's protocol. Immunoprecipitated DNA released from the cross-linked proteins was quantitated by ABI 7900 (Applied Biosystems) using the primers (Table S3) and was normalized to input DNA. Each experiment was performed in triplicate.

## ChIP sequencing

The construction of library was performed using NEBNext Ultra DNA Library Prep Kit for Illumina (New England Biolabs) according to the manufacturer's instructions. Briefly, the chipped DNA was ligated with adaptors. After purification, PCR was done with adaptor-ligated DNA and index primer for multiplexing sequencing. The library was purified using magnetic beads to remove all reaction components. The size of the library was assessed by Agilent 2100 Bioanalyzer (Agilent Technologies, Amstelveen). High-throughput sequencing was performed as paired-end 100-bp sequencing using NovaSeq 6000 (Illumina, Inc.). The results were analysed by Integrated Genome Browser 10.0.0 (bioviz.org). De novo motifs were identified from the STAT3 and c-SKI ChIP-seq–binding sites using HOMER de novo motif analysis.

## Statistics

Data were analysed using a two-tailed unpaired $t$ test. A $P$-value < 0.05 was considered to indicate statistical significance.

# Supplementary Information

# Acknowledgements

We thank Dr. Michael Sporn (Dartmouth Medical College, USA) and Dr. Lalage Wakefield (National Institutes of Health, USA) for helpful discussions and critical reading of the article. We thank Dr. Chuxia Deng (National Institutes of Health, USA) for Smad3$^{ex8/ex8}$ mice. Microarray and RNA-seq works benefited from data assembled by the ImmGen consortium. The project was funded by JSPS KAKENHI JP16K09908, NRF-2015R1A1A3A04001 051, NRF-2018R1A2B6009255, and NRF-2021R1F1A1061654 to M Mamura; JSPS KAKENHI JP17K15735, JP19K16699, NRF-2016R1D1A1B03931785, and NRF-2019R1F1A1063262 to J-H Yoon; JSPS KAKENHI JP18K15252, JP20K16368, WISET-2017-083, WISET-2018-737, and NRF-2018R1A6A3A01011885 to E Bae; NRF SRC 2017R1A5A1014560 to SH Park; and KHIDI-HI16C1501 to I-K Lee.

## Author Contributions

J-H Yoon: conceptualization, resources, data curation, formal analysis, funding acquisition, validation, investigation, visualization, methodology, project administration, and writing—original draft, review, and editing.
E Bae: data curation, formal analysis, funding acquisition, investigation, visualization, methodology, and writing—original draft, review, and editing.
Y Nagafuchi: data curation, formal analysis, and writing—original draft, review, and editing.
K Sudo: data curation, formal analysis, and writing—original draft.
JS Han: data curation, formal analysis, and writing—original draft.
SH Park: data curation, formal analysis, and writing—original draft.
S Nakae: data curation, formal analysis, and writing—original draft.
T Yamashita: data curation, formal analysis, and writing—original draft.
JH Ju: data curation, formal analysis, and writing—original draft.
I Matsumoto: data curation, formal analysis, and writing—original draft.
T Sumida: data curation, formal analysis, and writing—original draft.
K Miyazawa: resources, data curation, formal analysis, and writing—original draft, review, and editing.
M Kato: data curation, formal analysis, and writing—original draft.
M Kuroda: data curation, formal analysis, and writing—original draft.
I-K Lee: data curation, formal analysis, and writing—original draft.
K Fujio: data curation, formal analysis, and writing—original draft, review, and editing.
M Mamura: conceptualization, resources, data curation, formal analysis, supervision, funding acquisition, validation, investigation, visualization, methodology, project administration, and writing—original draft, review, and editing.

## Conflict of Interest Statement

The authors declare that they have no conflict of interest.

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
