## [Reviewer comments · Life Science Alliance]

Life Science Alliance

Repression of SMAD3 by STAT3 and c-Ski induces conventional dendritic cell differentiation

Jeong-Hwan Yoon, Eunjin Bae, Yasuo Nagafuchi, Katsuko Sudo, Jin Soo Han, Seok Park, Susumu Nakae, Tadashi Yamashita, Ji Ju, Isao Matsumoto, Takayuki Sumida, Keiji Miyazawa, Mitsuyasu Kato, Masahiko Kuroda, In-kyu Lee, Keishi Fujio, and Mizuko Mamura

DOI: <https://doi.org/10.26508/lsa.201900581>

Corresponding author(s): Mizuko Mamura, Kyungpook National University Hospital

Review Timeline:

Submission Date:	2019-10-21
Editorial Decision:	2019-11-27
Revision Received:	2024-05-31
Editorial Decision:	2024-06-07
Revision Received:	2024-06-17
Accepted:	2024-06-18

Transaction Report:

November 27, 2019

Re: Life Science Alliance manuscript #LSA-2019-00581-T

Prof. Mizuko Mamura
Tokyo Medical University/Kyungpook National University Hospital
Department of Molecular Pathology, Diversity Promotion Center/Biomedical Research Institute
6-1-1 Shinjuku, Shinjuku-ku
Tokyo/Daegu 160-8402
Japan

Dear Dr. Mamura,

Thank you for submitting your manuscript entitled "SMAD3 Determines Conventional versus Plasmacytoid Dendritic Cell Fates" to Life Science Alliance. The manuscript was assessed by expert reviewers, whose comments are appended to this letter.

As you will see, the reviewers appreciate your analyses and provide constructive input to further strengthen your manuscript. We would thus like to invite you to submit a revised version of your manuscript, addressing the individual points raised by the reviewers. This seems rather straightforward, but please do get in touch with us in case you would like to discuss individual revision points further.

To upload the revised version of your manuscript, please log in to your account: <https://lsa.msubmit.net/cgi-bin/main.plex>
You will be guided to complete the submission of your revised manuscript and to fill in all necessary information. Please get in touch in case you do not know or remember your login name.

Thank you for this interesting contribution to Life Science Alliance. We are looking forward to receiving your revised manuscript.

Sincerely,

-- Summary blurb (enter in submission system): A short text summarizing in a single sentence the study (max. 200 characters including spaces). This text is used in conjunction with the titles of papers, hence should be informative and complementary to

the title and running title. It should describe the context and significance of the findings for a general readership; it should be written in the present tense and refer to the work in the third person. Author names should not be mentioned.

B. MANUSCRIPT ORGANIZATION AND FORMATTING:

Reviewer #1 (Comments to the Authors (Required)):

Review Yoon et al.

This is a very interesting and comprehensive analysis of the regulation of cDC/pDC differentiation from progenitor cells by Smad3 which adds to our understanding of this complicated process. I think the manuscript is quite thoroughly done and quite well written, but it needs some revision.

Major points:

Abstract: In my opinion it is overstated to say Smad3 is the pivotal determinant for cDC and pDC differentiation (last sentence). Better to say "Smad3 is an important regulator..." or such.

Figure 2B: The total percentage of CD11c⁺ MHCII⁺ cells generated in the BM cultures on day 7 seems to be rather low. Also the frequency of pDCs generated in Flt3L cultures was quite low (< 10 % in the control conditions). It would therefore be important to know if the transfection affected the differentiation of the BM cells in culture with Flt3L or GM-CSF/IL-4 and state this in the results text. Also, it should be clarified if gating on transfected cells was performed or not.

Figure 2D: The authors conclude from this data that "Smad3-dependent TGF-beta signaling induces pDC differentiation while (it) inhibits cDCs differentiation". This cannot be concluded from the data, because TGF-beta abrogates DC differentiation entirely in Flt3L cultured BM cells (3.6 % CD11c⁺/MHCII⁺, 0.5 % pDC) and this global effect is Smad3-dependent. The pDC-inducing effect of TGF-beta can only be seen at the lower doses as shown in figure 4. So this conclusion can only be drawn later in the manuscript.

Figure 3: This figure is too crowded to be easily understandable. I recommend to move some of the dot plots into the supplement and show the summarized data only.

In figure 3 cell numbers in BM or cell numbers in CD11c⁺ gate are shown. How were these cell numbers determined? Why is the frequency of the different DC subtypes within CD11c⁺ cells not shown (in fig. 3E)?

Figure 3A does not include the CLP and Ly6D⁺ Siglec H⁺ lymphoid precursors of pDC. Since these were shown to generate the majority of pDCs (Rodrigues et al Nat Immunol 2018, Dress et al Nat Immunol 2019), they should also be investigated.

Figure 4A: Also here the Ly6D⁺ and Ly6D⁺ Siglec H⁺ pre-pDC should be investigated if sufficient amount of cells can be isolated.

Figure 5A: It is problematic to say that PDCA1⁺ CD11b⁻ cells generated in Flt3L culture are truly pDCs - they could be precursors of conventional DCs, clear B220 expression and/or Ly6D must be shown in addition.

Figure 5E: The occupancy of the Smad3 promoter is higher with anti-H3K4me3 than with Anti-H3K27me3 ChIP. This is not stated corrected in the results text. It would help to briefly explain the role of the different acetyl and trimethyl histones (also for figure 6 E).

Discussion: In the light of recent publication on the (exclusive?) lymphoid progenitor origin of pDCs (Rodrigues et al Nat Immunol 2018, Dress et al Nat Immunol 2019), it would be important to clarify in which precursor cell type the pDC cell fate is enforced by Smad3. Does this happen at the SiglecH⁺ Ly6D⁺ pre-pDC stage or earlier in the Ly6D⁺ lymphoid progenitor of pDCs? Or does this regulation happen in a true "common" progenitor/precursors of pDCs and cDCs? This should be discussed and the frequency and number of these precursors should be investigated (see above comments on fig. 3A and 4A).

Minor:

Show gating strategy for all the populations ex vivo and from BM cultures in the supplement. Show transfection efficiency with siRNA and overexpressed proteins.

Figure 1 B lower panels: PDCA1 written in yellow - not visible in printout

Legend to Figure 1: say what is shown exactly? Mean of 5 independent experiments or of one experiment with several replicates? This should be clarified in all the legends.

Grammar errors, e.g.: "SMAD3-mediated TGF-beta signaling induces pDC while inhibiting cDC differentiation."

Page 7, sentence 2 and 3: Better use "but" instead of "while"

Page 10 (and several other similar sentences): "Thus, Smad3-mediated low-dose TGFb signaling induces pDC-related genes while it represses cDC-related genes." The "it" is missing after "while". Same on page 8.

Reviewer #2 (Comments to the Authors (Required)):

Yoon and colleagues investigate the importance of TGFb signalling and specifically SMAD3 in dendritic cell fate determination. They show in mouse models and in vitro cell culture assays that SMAD3 is essential for pDC development and that SMAD3 is specifically repressed during cDC differentiation. They show that STAT3, downstream of FLT3, acts with the corepressor c-SKI to repress SMAD3 in cDCs. In contrast TGFb signalling is shown to promote SMAD3 and thus reinforce the pDC pathway.

Overall this is a thorough and generally convincing study. A strength is the large number of molecular and cellular assays that are mostly supportive of the authors model (Figure 8). Perhaps a limitation of the study is that the molecular mechanisms are exclusively centred on the proximal promoter of SMAD3 and depend on the use of a SMAD3 luciferase reporter construct. ChIPseq for SMADS (2 or 4 and 3) in cDC and pDC would potentially provide a lot of support for the authors molecular mechanisms more generally.

Some of the data presented here, disagree with previous studies on TGFb in DCs, particularly from the Zenke lab. I find the current data and the authors discussion of the potential differences between the approaches reasonable.

Specific comments.

Figure 2. Some of the FACS staining in this figure appears sub-optimal. When "CD11c+ gated cells" is indicated does this include all CD11c+ or is the gating limited to the CD11c-hi cells that also express high MHCII ? (eg in panel D). Including a lot of CD11c-low non-DCs will skew the quantitation in some conditions. In addition, the staining for CD24 (a cDC1 marker) is very weak and likely to be underestimating the cDC1 numbers. A similar weakness is in the PDCA1/bst2 staining.

Figure3 contains too much information. It is difficult for the reader to follow. I suggest thinning out the figure by moving some to Suppl figures.

Figure 3B. The upper FACS plots for panel B (MDP) are duplicated in the lower panels (CDP). I assume this is an error. Also again the gating is very tight. It appears that a clear CD135+CD115+ population is seen in the +/+ that is absent in the -/-, yet the use of such a tight gate incorporates many cells that are not truly CD135+CD115+. The authors need to replace the duplicated plots and provide more appropriate gating on the MDP and CDP.

Figure 4B-E is interpreted to show that TGFb induced pDC genes and represses cDC genes. As SMAD3 overexpression (or loss) or TGFb exposure influences cDC/pDC developmental ratios, I feel that the QPCR data is simply reflecting the changes in cellular makeup of the cultures. Much more targeted experiments, such as short time points and SMAD ChIPseq are needed to make the conclusion that TGFb/SMAD3 is influencing these genes directly. I feel these conclusions, as stated in the heading for the figure legend and the results need to be more conservative.

Figure 4/5. The dose titration experiments on TGFb inducing SMAD3 are conflicting. For example figure 4C shows a different dose response curve for Smad3 mRNA to TGFb than the promoter construct in figure 5D. Figure 4C generally agrees with the IHC experiments in Figure 5C. The authors suggest that the protein difference might be due to ligand dependent degradation, but this doesn't explain the mRNA differences. It is also plausible that the promoter construct is missing some negative regulatory sequences or that high doses of TGFb are simply uninformative.

Reviewer #3 (Comments to the Authors (Required)):

The authors report that Smad3 promotes pDC development and blocks cDC development. pDC development is induced by Smad3 overexpression and is lost in Smad3^{-/-} mice. This differential effect of Smad3 on pDC and cDC starts earlier in DC lineage at CDP stage. Via overexpression experiments the authors come to a mechanism explaining the Smad3 effect: in pDC development there is an autoregulatory mechanism that drives expression of Smad3, whereas in cDC development expression of Smad3 is repressed by other signaling molecules like Stat3.

This manuscript adds to the understanding how cell signaling pathways instruct differentiation of dendritic cells. The conclusions

drawn by the authors are in most cases supported by the data, however, I have some concerns:

- The in vitro BM differentiation appears poor to me. In Flt3L BMC cultures 80-90% of cells should be CD11c+ after 8-9 days. Here it seems max 50%. Why is that? Were these cells analyzed too early? pDC, cDC1 and cDC2 develop with different kinetics so the relative distribution of these subsets depends on timing of the analysis. It is not clear to me if culture time was always the same. Methods say cells were cultured for 2-9 days.
- For some markers (eg SiglecH, B220) the signal intensity in flow cytometry is so low that it is very hard to clearly distinguish positive from negative cells, which raises questions about the quantification based on these markers. Can this be improved?
- In several experiments genes were overexpressed via transient transfection. I seem to understand that BM cells were transfected before they were induced to differentiate in vitro. How was transfection efficiency determined? In some cases cotransfection of 5 constructs was done. How was it verified that cells got transfected with all constructs? Does overexpression last until the point of analysis?

I put here my comments with the individual figures:

Figure 1:

- Nomenclature of cell populations is not consistent between figure and text, which makes it hard to follow. Eg CD11chi cDC in text is this the same as CD11c+CD11b+ in the figure? This would be cDC2 then?
- Panel A: The in vitro cultured cells, are they sorted for DC subsets? If not, what is the relevance of the Smad3 expression, it would be an average of pDC and cDC.
- Panel B: too small to see dots in the nucleus. Moreover, how were these dots quantified? Are cells spun down on glass slides? If yes, how can one be sure that 'dots' are nuclear? Might be better to show only Smad3. The yellow color is confusing as it might come from pdca1 but also from red/green overlap.
- Bar chart should be panel C.

Figure 2:

- Transfection was done with crude bone marrow cells or after lineage depletion? What was the efficiency?
- Smad3 overexpression inhibits DC differentiation almost completely. Thus, although what is left might have pDC characteristics, the absolute pDC number might be much less compared to control. If that is the case I find it hard to claim that 'forced expression of Smad3 resulted in significant increase in PDCA1+B220+ pDC'. What are the absolute cell numbers here?
- pDC would be B220+ PDCA1+, it is easier if only that gate is shown instead of quadrants.
- 5 ng/ml TGFb1 is a high concentration and will kill most cells I guess (rather than inhibit differentiation). What are absolute cell numbers with this high dose TGFb1?
- PDCA1+ B220+ pDC in GMCSF/IL4 culture (supplemental figure) are not convincing.

Figure 3:

- The entire figure should be reduced in complexity. Instead of using quadrants please use single gates where possible, the figure is already complex enough.
- Nomenclature of cell populations in text and figure is not consistent.
- In panel B the contour plots for CD117hi and CD117int look identical. Please check.

Figure 4:

Gene expression by Flt3l cultured, transfected bone marrow cells.

- As figure 2B says that Smad3 overexpression inhibits DC differentiation, what cells are left and what is the meaning of the gene expression?
- It is stated that 'forced expression of Smad3 significantly upregulates the mRNA expression of pDC related genes'. Isn't this 'upregulation' just reflecting the fact that the few cells that are left are pDC-like, rather than that Smad3 would directly target pDC genes? To claim that Smad3 directly regulates pDC genes, ChIP for Smad3 should be done followed by analysis for pDC gene loci.

Figure 5:

- The 'pDC' in GMCSF/IL4 cultures treated with a bit of TGFb, how solid is it that they are pDC? Are there other pDC markers used, do these cells express other typical pDC genes (function related for example)? Or is it just an upregulation of PDCA1?
- the green/red/yellow combination in C is confusing, as yellow can be PDCA1 or green/red overlap (see also Figure 1). Why are not all cells Smad3+ or P-Smad3+? Why is there little difference in P-Smad3+ cells between TGFb treated (low dose) and untreated cells?
- In the ChIP experiments, cells were transfected with the Smad3-luciferase reporter. Why?
- Panel E and F are very poorly described in the text.

Figure 6:

- Cells are transfected with many constructs. How is it verified that cells have received all?

Figure 7:

- For the IF, 2 different proteins are red-colored, how does this work? The quantification of the red color in panel A is hard to understand. Red signal is more intense in BM CD11c+CD11b+ compared to BMDC however the bar diagram suggests equal

PLA signal in these cells. How can that be? Or is the red color for the BMDC somehow masked in IF?

General comments:

- The figures are too complex, they are overloaded with data and often too poorly described to follow the author's reasoning. If only the essential data are shown readability of the manuscript would improve.
- Flow cytometry data are all contour plots with quadrants. In most cases however there is only 1 population of interest per plot. Showing this one gate instead of quadrants again would decrease complexity a lot and improve readability.
- Statistics in bar diagrams: not clear what is compared to what and why.

Responses to the Reviewer

Reviewer #1

First of all, we extend our deepest gratitude to your exceptional consideration and understanding for our difficulties during the pandemic. We deeply appreciate your constructive comments on our manuscript for improvement. Other reviewers instructed to perform ChIP-seq and to reduce the data to avoid complexity and to improve readability. Because of the increased data volume as a result of responding to the comments, we have decided to focus on conventional DC differentiation with the revised title: **Repression of SMAD3 by STAT3 and c-SKI induces conventional dendritic cell differentiation**. We have invited Dr. Yasuo Nagafuchi and Dr. Keishi Fujio, Department of Allergy and Rheumatology, University of Tokyo as coauthors for their expertise in bioinformatics. We have incorporated the recent advancement in the research field as well as analysis of the open data sources (Supplementary Figure 2) to support our findings. Please find our responses (revised parts are underlined) to your general comments and the specific comments on cDCs (displayed in bold).

Review Yoon et al.

This is a very interesting and comprehensive analysis of the regulation of cDC/pDC differentiation from progenitor cells by Smad3 which adds to our understanding of this complicated process. I think the manuscript is quite thoroughly done and quite well written, but it needs some revision.

Major points:

Abstract: In my opinion it is overstated to say Smad3 is the pivotal determinant for cDC and pDC differentiation (last sentence). Better to say "Smad3 is an important regulator..." or such.

We eliminated subjective expression and revised the last sentence in the Abstract; These data indicate that STAT3 and c-Ski induce cDC differentiation by repressing SMAD3: the repressor of the cDC-related genes during the developmental stage between MDP and CD115⁺ CDP.

Figure 2B: The total percentage of CD11c⁺ MHCII⁺ cells generated in the BM cultures on day 7 seems to be rather low.

Please refer to Figure 3B in the revised manuscript. We reanalyzed the data with less tight gating procedures than those in the original version.

It would therefore be important to know if the transfection affected the differentiation of the BM cells in culture with FLt3L or GMCSF/IL-4 and state this in the results text.

Following the reviewer's comment, we stated the effect of transfection on BMDC differentiation in the Results. Please refer to page 9, line 197-199 (Transfection of pcDNA or control siRNA reduced the proportions of the cells highly positive for MHCII and CD11c, however, net percentages of MHCII⁺CD11c⁺ were not altered by transfection (Fig 3B and 3D).

Also, it should be clarified if gating on transfected cells was performed or not.

Please refer to page 8, line 185-188 (Expression levels of Flag-tagged SMAD3 were confirmed by flowcytometry (Fig S4A) and immunocytochemistry using PLA (Fig S4B).

SMAD3 mRNA in the transfected BMDCs were confirmed by quantitative RT-PCR (RT-qPCR; Fig S4C).

Page 14, line 328-329 (Expression levels of Flag-tagged c-SKI and mutants were confirmed by flowcytometry (Fig S6C) and immunocytochemistry using PLA (Fig S6D).)

page 24, line 552-557 (Knockdown efficiencies by siRNA were confirmed by RT-qPCR. Transfection efficiencies of Flag-tagged plasmids were confirmed by detection of Flag using flowcytometry and immunocytochemistry by PLA, with approximately 55-60 % and 80% of transfection efficiencies, respectively.)

Figure 3: This figure is too crowded to be easily understandable. I recommend to move some of the dot plots into the supplement and show the summarized data only.

Please find Figure 2 and Supplementary Figure 3B in the revised manuscript.

In figure 3 cell numbers in BM or cell numbers in CD11c+ gate are shown. How were these cell numbers determined?

Please refer to revised Figure 2. Please find page 20, line 451-452 and page 21, line 484-485 (Cell numbers were counted using CountessTM 3 automated cell counter (AMQAX2000, ThermoFisher Scientific).)

Minor:

Show gating strategy for all the populations ex vivo and from BM cultures in the supplement.

Please find Supplementary Figure 3A (ex vivo) and 4E (BM cultures) in the revised manuscript.

Show transfection efficiency with siRNA and overexpressed proteins.

Please find the mRNA expression in the siRNA-transfected cells in Supplementary Figure 4C and 6A in the revised manuscript. We detected FLAG for the tagged plasmids using flowcytometry and immunocytochemistry with proximity ligation assay (Supplementary Figure 4A, 4B, 6C and 6D).

Legend to Figure 1: say what is shown exactly? Mean of 5 independent experiments or of one experiment with several replicates? This should be clarified in all the legends.

We have elaborated the indicated points in the figure legends in the revised manuscript.

Responses to the Reviewer

Reviewer #2

First of all, we extend our deepest gratitude to your exceptional consideration and understanding for our difficulties during the pandemic. We deeply appreciate your constructive comments on our manuscript for improvement. Your precious comments on ChIP-seq and other reviewer's instructions to reduce the complexity to improve readability directed us to focus on conventional DC differentiation with the revised title: **Repression of SMAD3 by STAT3 and c-SKI induces conventional dendritic cell differentiation**. Therefore, we performed ChIP-seq to determine the binding sites of STAT3 and c-Ski and their chromatin states in the *Smad3* promoter region. We have invited Dr. Yasuo Nagafuchi and Dr. Keishi Fujio, Department of Allergy and Rheumatology, University of Tokyo as coauthors for their expertise in bioinformatics. We have incorporated the recent advancement in the research field as well as analysis of the open data sources (Supplementary Figure 2) to support our findings. Please find our responses (revised parts are underlined) to your general comments and the specific comments on cDCs (displayed in bold).

Overall this is a thorough and generally convincing study. A strength is the large number of molecular and cellular assays that are mostly supportive of the authors model (Figure 8). Perhaps a limitation of the study is that the molecular mechanisms are exclusively centred on the proximal promoter of SMAD3 and depend on the use of a SMAD3 luciferase reporter construct. ChIPseq for SMADS (2 or 4 and 3) in cDC and pDC would potentially provide a lot of support for the authors molecular mechanisms more generally.

We appreciate your instruction to perform ChIP-seq. To examine the repression of SMAD3 by STAT3 and c-Ski in cDCs, we performed ChIP-seq to identify enriched loci of STAT3 and c-Ski together with histone modification states in the coding and flanking regions of SMAD3 in FLT3L-induced BMDCs. Please refer to Figure 5E and page 12, line 267-284.

Some of the data presented here, disagree with previous studies on TGFb in DCs, particularly from the Zenke lab. I find the current data and the authors discussion of the potential differences between the approaches reasonable.

Figure 2. Some of the FACS staining in this figure appears sub-optimal. When "CD11c+ gated cells" is indicated does this include all CD11c+ or is the gating limited to the CD11c-hi cells that also express high MHCII ? (eg in panel D). Including a lot of CD11c-low non-DCs will skew the quantitation in some conditions.

Please refer to revised Figure 3. We have corrected the description to CD11c⁺MHCII⁺ gate.

In addition, the staining for CD24 (a cDC1 marker) is very weak and likely to be underestimating the cDC1 numbers.

Percentages of CD24⁺ cells in CD11c⁺MHCII⁺ gated FLT-3L-induced BMDCs are similar to the published data such as Purvis, Harriet A., et al. "Phosphatase PTPN22 regulates dendritic cell homeostasis and cdc2 dependent T cell responses." *Frontiers in immunology* 11 (2020): 376.

Figure3 contains too much information. It is difficult for the reader to follow. I suggest thinning out the figure by moving some to Suppl figures.

Please find Figure 2 and Supplementary Figure 3B in the revised manuscript.

Figure 3B. The upper FACS plots for panel B (MDP) are duplicated in the lower panels (CDP). I assume this is an error. Also again the gating is very tight. It appears that a clear CD135+CD115+ population is seen in the +/+ that is absent in the -/-, yet the use of such a tight gate incorporates many cells that are not truly CD135+CD115+. The authors need to replace the duplicated plots and provide more appropriate gating on the MDP and CDP.

We have corrected the duplicated contour plots. Please refer to Figure 2B in the revised manuscript.

Figure 4B-E is interpreted to show that TGFb represses cDC genes. As SMAD3 overexpression (or loss) or TGFb exposure influences cDC developmental ratios, I feel that the QPCR data is simply reflecting the changes in cellular makeup of the cultures. Much more targeted experiments, such as short time points and SMAD ChIPseq are needed to make the conclusion that TGFB/SMAD3 is influencing these genes directly.

Please refer to Figure 4 A-D in the revised manuscript. We added shorter time point, day 1 for FLT3L-induced BMDC.

We performed ChIP-seq to identify enriched loci of STAT3 and c-Ski together with histone modification states in the coding and flanking regions of SMAD3 in FLT3L-induced BMDCs as stated above. Please refer to Figure 5E and page 12, line 267-284.

I feel these conclusions, as stated in the heading for the figure legend and the results need to be more conservative.

We have revised the heading for the figure legend and the results (page 9, line 209 and page 44, line 998, SMAD3-mediated TGF- \$\beta\$ signaling downregulates cDC-related genes).

Responses to the Reviewer

Reviewer #3

First of all, we extend our deepest gratitude to your exceptional consideration and understanding for our difficulties during the pandemic. We deeply appreciate your constructive comments on our manuscript for improvement. Your precious comments on reducing data volume for readability together with an instruction to perform ChIP-seq by other reviewer directed us to focus on conventional DC differentiation for revision. We have invited Dr. Yasuo Nagafuchi and Dr. Keishi Fujio, Department of Allergy and Rheumatology, University of Tokyo as coauthors for their expertise in bioinformatics. We have incorporated the recent advancement in the research field as well as the open data sources (Supplementary Figure 2) to support our findings. Please find our responses (revised parts are underlined) to your general comments and the specific comments on cDCs (displayed in bold).

This manuscript adds to the understanding how cell signaling pathways instruct differentiation of dendritic cells. The conclusions drawn by the authors are in most cases supported by the data, however, I have some concerns:

- The in vitro BM differentiation appears poor to me. In Flt3L BMC cultures 80-90% of cells should be CD11c+ after 8-9 days. Here it seems max 50%. Why is that? Were these cells analyzed too early?

Please refer to page 20, line 459. We cultured BM cells for 7-8 days.

pDC, cDC1 and cDC2 develop with different kinetics so the relative distribution of these subsets depends on timing of the analysis. It is not clear to me if culture time was always the same. Methods say cells were cultured for 2-9 days.

We have revised the Materials and Methods, page 20, line 451-460. For time course analyses in Figure 4A and 4B, we cultured BMDCs for 1-8 days.

- In several experiments genes were overexpressed via transient transfection. I seem to understand that BM cells were transfected before they were induced to differentiate in vitro. How was transfection efficiency determined? In some cases cotransfection of 5 constructs was done. How was it verified that cells got transfected with all constructs? Does overexpression last until the point of analysis?

Please refer to

page 8, line 185-188 (Expression levels of Flag-tagged SMAD3 were confirmed by flowcytometry (Fig S4A) and immunocytochemistry using PLA (Fig S4B). SMAD3 mRNA in the transfected BMDCs were confirmed by quantitative RT-PCR (RT-qPCR; Fig S4C).)

Page 14, line 328-329 (Expression levels of Flag-tagged c-SKI and mutants were confirmed by flowcytometry (Fig S6C) and immunocytochemistry using PLA (Fig S6D).)

page 24, line 552-556 (Knockdown efficiencies by siRNA were confirmed by RT-qPCR. Transfection efficiencies of Flag-tagged plasmids were confirmed by detection of Flag using flowcytometry and immunocytochemistry by PLA, with approximately 55-60 % and 75-80% of transfection efficiencies, respectively.)

For luciferase reporter assay, one up to 6 constructs were transfected. We used dual luciferase reporter assay, in which the activity of the transcriptional regulatory elements is evaluated by Firefly luciferase, whereas Renilla luciferase acts as an internal transfection control in the same sample.

I put here my comments with the individual figures:

Figure 1:

- Nomenclature of cell populations is not consistent between figure and text, which makes it hard to follow. Eg CD11chi cDC in text is this the same as CD11c+CD11b+ in the figure? This would be cDC2 then?

We have matched the nomenclature of cell populations in figures and text in the revised manuscript.

- Panel A: The in vitro cultured cells, are they sorted for DC subsets? If not, what is the relevance of the Smad3 expression, it would be an average of pDC and cDC.

As you indicated, BMDCs were not sorted. GM-CSF/IL-4-induced BMDCs do not develop pDC, therefore, we compared the *Smad3* mRNA expression in FLT3L-induced BMDCs and in GM-CSF/IL-4-induced BMDCs

- Panel B: too small to see dots in the nucleus. Moreover, how were these dots quantified? Are cells spun down on glass slides? If yes, how can one be sure that 'dots' are nuclear?

We spun down the BM cells and splenic DCs on glass slides using a cytospin centrifuge (Thermoscientific). Expression of SMAD2 and SMAD3 proteins were detected using proximity ligation assay (PLA), in which each PLA signal is composed of ~1000 bound fluorescent probes that appear as a distinct dot, not like smears appeared in conventional immunocytochemistry (Sigma-Aldrich). We added the paper by the scientists who developed the technique: Söderberg Ola, et al. Direct observation of individual endogenous protein complexes in situ by proximity ligation. *Nat Methods* 3: 995-1000 in References for detailed technical information. Red PLA signals in 10 images from each experiment were calculated using BlobFinder software as described in Immunocytochemistry, Material and Methods section (page 23).

We observed the PLA-stained specimens by confocal microscopies: LSM700 and LSM900 with 374 nm axial resolution at 550 nm (page 23), which are able to detect cellular localization of proteins (eg. Lönn, P., Al-Amin, R. A., Doulabi, E. M., Heldin, J., Gallini, R., Björkesten, J., ... & Landegren, U. (2021). Image-based high-throughput mapping of TGF- β -induced phosphocomplexes at a single-cell level. *Communications Biology*, 4(1), 1284).

- Bar chart should be panel C.

We labeled the bar graphs of PLA data as Figure 1C in the revised manuscript.

Figure 2:

Please refer to Figure 3 in the revised manuscript.

- Transfection was done with crude bone marrow cells or after lineage depletion? What was the efficiency?

Transfection was done with crude bone marrow cells (Materials and Methods, page 24 line 542 to 557).

Please refer to page 8, line 185-188 (Expression levels of Flag-tagged SMAD3 were confirmed by flowcytometry (Fig S4A) and immunocytochemistry using PLA (Fig S4B). SMAD3 mRNA in the transfected BMDCs were confirmed by quantitative RT-PCR (RT-qPCR; Fig S4C).

page 24, line 552-556 (Knockdown efficiencies by siRNA were confirmed by RT-qPCR. Transfection efficiencies of Flag-tagged plasmids were confirmed by detection of Flag using flowcytometry and immunocytochemistry by PLA, with approximately 55-60 % and 75-80% of transfection efficiencies, respectively).

- Smad3 overexpression inhibits DC differentiation almost completely. What are the absolute cell numbers here?

Please refer to Supplementary Figure 4F.

- 5 ng/ml TGFb1 is a high concentration and will kill most cells I guess (rather than inhibit differentiation). What are absolute cell numbers with this high dose TGFb1?

Please refer to Supplementary Figure 4G. TGF- β 1 inhibited cDC differentiation, while increasing cell viability.

Figure 3:

Please refer to Figure 2 in the revised manuscript.

- The entire figure should be reduced in complexity.

Please refer to Figure 2 in the revised manuscript.

- Nomenclature of cell populations in text and figure is not consistent.

We have matched nomenclature of cell populations in text and figure.

- In panel B the contour plots for CD117^{hi} and CD117^{int} look identical. Please check.

We have corrected the duplicated contour plots. Please refer to Figure 2B in the revised manuscript.

Figure 4:

Gene expression by Flt3l cultured, transfected bone marrow cells.

- As figure 2B says that Smad3 overexpression inhibits DC differentiation, what cells are left and what is the meaning of the gene expression?

Please refer to Figure 4 in the revised manuscript. Overexpression of SMAD3 inhibited cDC differentiation (Figure 3B), while keeping the cells in progenitor stages (Figure 3A). Therefore, we examined the effect of SMAD3 overexpression on the expression of cDC-related genes.

Figure 5:

- In the ChIP experiments, cells were transfected with the Smad3-luciferase reporter. Why?

I am afraid that this is misreading. We used SMAD3-luciferase reporter constructs for reporter assays, but not for ChIP experiments. We revised the explanation to avoid misunderstanding (Figure 5H and I, page 12, line 285-288).

Figure 6:

- Cells are transfected with many constructs. How is it verified that cells have received all?

Please refer to Figure 5A-D in the revised manuscript. We used dual luciferase reporter assay, in which the activity of the transcriptional regulatory elements is evaluated by Firefly luciferase, whereas Renilla luciferase acts as an internal transfection control in the same sample as stated above in the response to your general comments.

Figure 7:

- For the IF, 2 different proteins are red-colored, how does this work? The quantification of the red color in panel A is hard to understand.

Please refer to revised Figure 6. We evaluated the protein-protein interaction using Duolink proximity ligation assay (Sigma-Aldrich). Red dot signals represent protein-protein interactions in the nucleus and cytoplasm, which were quantified using BlobFinder software.

We added the paper by the scientists who developed the technique: Söderberg, Ola, et al. Direct observation of individual endogenous protein complexes in situ by proximity ligation. *Nat Methods* 3: 995-1000 in References for detailed technical information as stated above.

Red signal is more intense in BM CD11c+CD11b+ compared to BMDC however the bar diagram suggests equal PLA signal in these cells. How can that be? Or is the red color for the BMDC somehow masked in IF?

Red dots can be masked by green colors through megascopic observation. Therefore, we calculated the red PLA signals in nucleus and cytoplasm using BlobFinder software.

General comments:

- The figures are too complex, they are overloaded with data and often too poorly described to follow the author's reasoning. If only the essential data are shown readability of the manuscript would improve.

We appreciate the reviewer's comment to avoid data overload for readability and focused on cDC differentiation in the revised manuscript.

- Flow cytometry data are all contour plots with quadrants. In most cases however there is only 1 population of interest per plot. Showing this one gate instead of quadrants again would decrease complexity a lot and improve readability.

As the reviewer indicated, we revised contour plots by single gate in Figure 2, 3 and 7.

- Statistics in bar diagrams: not clear what is compared to what and why.

We changed and adjust *P* values of statistical graphs.

June 7, 2024

RE: Life Science Alliance Manuscript #LSA-2019-00581-TR

Prof. Mizuko Mamura
Kyungpook National University Hospital
Biomedical Research Institute
135 Dongduk-ro Jung-gu
Daegu 41944
Korea, Republic of (South Korea)

Dear Dr. Mamura,

Thank you for submitting your revised manuscript entitled "Repression of SMAD3 by STAT3 and c-Ski induces conventional dendritic cell differentiation". We would be happy to publish your paper in Life Science Alliance pending final revisions necessary to meet our formatting guidelines.

- please be sure that the authorship listing and order is correct
- please upload your Table in editable .doc or Excel format
- please add the Twitter handle of your host institute/organization as well as your own or/and one of the authors in our system
- the contributions selected for Isao Matsumoto and Takayuki Sumida do not qualify them for authorship. Please either update the contributions in our system and the Author Contributions section of the manuscript or let us know if the authors need to be removed (and added eventually to the Acknowledgments section)
- since Figure 8 is a Graphical Abstract, please upload it with the file designation "Graphical abstract" and remove it from the figure legends
- please add your main, supplementary figure, and table legends to the main manuscript text after the references section
- please remove legends from the supplementary figures; their legends should only appear in the manuscript file

FIGURE CHECKS:

- please add sizes next to the blots in Figure S1A

A. FINAL FILES:

B. MANUSCRIPT ORGANIZATION AND FORMATTING:

Sincerely,

June 18, 2024

RE: Life Science Alliance Manuscript #LSA-2019-00581-TRR

Prof. Mizuko Mamura
Kyungpook National University Hospital
Biomedical Research Institute
135 Dongduk-ro Jung-gu
Daegu 41944
Korea, Republic of (South Korea)

Dear Dr. Mamura,

Thank you for submitting your Research Article entitled "Repression of SMAD3 by STAT3 and c-Ski induces conventional dendritic cell differentiation". It is a pleasure to let you know that your manuscript is now accepted for publication in Life Science Alliance. Congratulations on this interesting work.

DISTRIBUTION OF MATERIALS:

Again, congratulations on a very nice paper. I hope you found the review process to be constructive and are pleased with how the manuscript was handled editorially. We look forward to future exciting submissions from your lab.

Sincerely,
